# Overview and New Insights into the Metabolic Syndrome: Risk Factors and Emerging Variables in the Development of Type 2 Diabetes and Cerebrocardiovascular Disease

**DOI:** 10.3390/medicina59030561

**Published:** 2023-03-13

**Authors:** Melvin R. Hayden

**Affiliations:** Department of Internal Medicine, Endocrinology Diabetes and Metabolism, Diabetes and Cardiovascular Disease Center, University of Missouri School of Medicine, One Hospital Drive, Columbia, MO 65211, USA; mrh29pete@gmail.com

**Keywords:** endothelial dysfunction, exosomes, hyperinsulinemia, hyperglycemia, hyperlipidemia, hypertension, insulin resistance, leptin resistance, metainflammation, miRNAs

## Abstract

Metabolic syndrome (MetS) is considered a metabolic disorder that has been steadily increasing globally and seems to parallel the increasing prevalence of obesity. It consists of a cluster of risk factors which traditionally includes obesity and hyperlipidemia, hyperinsulinemia, hypertension, and hyperglycemia. These four core risk factors are associated with insulin resistance (IR) and, importantly, the MetS is known to increase the risk for developing cerebrocardiovascular disease and type 2 diabetes mellitus. The MetS had its early origins in IR and syndrome X. It has undergone numerous name changes, with additional risk factors and variables being added over the years; however, it has remained as the MetS worldwide for the past three decades. This overview continues to add novel insights to the MetS and suggests that leptin resistance with hyperleptinemia, aberrant mitochondrial stress and reactive oxygen species (ROS), impaired folate-mediated one-carbon metabolism with hyperhomocysteinemia, vascular stiffening, microalbuminuria, and visceral adipose tissues extracellular vesicle exosomes be added to the list of associated variables. Notably, the role of a dysfunctional and activated endothelium and deficient nitric oxide bioavailability along with a dysfunctional and attenuated endothelial glycocalyx, vascular inflammation, systemic metainflammation, and the important role of ROS and reactive species interactome are discussed. With new insights and knowledge regarding the MetS comes the possibility of new findings through further research.

## 1. Introduction

Metabolic syndrome (MetS) is a cluster of multiple risk factors and variables that are associated with an increased risk for the development of atherosclerotic cardiovascular disease and type 2 diabetes mellitus (T2DM) [1,2]. In this review, cardiovascular disease will also include cerebrovascular disease or, more specifically, the two combined as cerebrocardiovascular disease (CCVD), since the vasculature system, heart, and brain are all involved in the clinical utility of the MetS. The core underlying risk factors include: (1) obesity (central or visceral obesity), systemic insulin resistance (IR) with hyperinsulinemia, hyperamylinemia, hyperleptinemia, and leptin resistance (LR); (2) hyperlipidemia with atherogenic dyslipidemia typified by elevated very-low-density lipoproteins or triglycerides, elevated small dense low-density lipoproteins-cholesterol, and decreased high-density lipoproteins-cholesterol; (3) essential hypertension (HTN); and (4) hyperglycemia with or without manifest T2DM [1,2].

Since the early days in 1923 (Kylin) [3] and 1939 (Himsworth and Kerr) [4], it was clinically obvious to many that certain CCVD risk factors and variables tended to cluster or co-occur, and that this clustering phenomenon shared multiple underlying causes, pathobiological mechanisms, and features. Following Reaven’s definition of syndrome X in 1988 [5], the National Cholesterol Education Panel Adult Treatment Panel III (NCEP ATP III) identified and characterized the MetS as being comprised of five clinical and laboratory screening criteria: (1) abdominal obesity by waist circumference (males >100 cm/>40 inches and females >88 cm/>35 inches reflecting increased visceral adiposity and possible IR; (2) triglycerides ≥150 mg/dL; (3) HDL-cholesterol <40 mg/dL in males and <50 mg/dL females reflecting atherogenic dyslipidemia and hyperlipidemia; (4) blood pressure ≥130/≥85 mm Hg reflecting HTN; (5) fasting glucose ≥110 mg/dL and the newer American Diabetes Association cut point of ≥100 mg/dL in 2002. Importantly, any three of the previously listed five criteria qualify individuals for the diagnosis of the MetS [1]. Importantly, note that in the NCEPT ATP III guidelines increased triglycerides (2) are separated from low HDL-cholesterol (3), reflecting atherogenic dyslipidemia, and that is why NCEP ATP III differs somewhat from the earlier mentioned four core risk factors in the preceding paragraph.

Since the MetS was first described, it has been described by many different major global organizations/associations and given many different names with multiple risk factors and variables added to what could now be described as the MetS reloaded. In its simplest form, the MetS consists of multiple clinically relevant risk factors and associated variables that intersect in the development atherosclerotic CCVD and T2DM (Figure 1).

T2DM and CCVD may also be associated with neurodegenerative diseases such as late-onset Alzheimer’s disease (LOAD), Parkinson’s disease, vascular dementia (VaD), mixed co-occurrence dementia, and diabetic encephalopathy—cognopathy [6,7,8,9]. Notably, the combination of CCVD and chronic kidney disease together comprise the heart–brain-kidney (HBK) axis in the MetS reloaded [10,11,12,13]. 

Initially, hyperinsulinemia is protective and a compensatory response to nutrient excess and obesity to control glucose elevation. When hyperinsulinemia is sustained chronically, it becomes detrimental and results in accelerated atherogenesis with the development of CCVD (Figure 2) [11,12,14].

aMt adds even further strength to understanding the important role of the MetS reloaded because Mt dysfunction plays an important role in systemic and brain insulin resistance (BIR), HTN, and T2DM. Notably, Reaven added the risk factors hyperuricemia and increased plasminogen activator-1 (PAI-1), and these were incorporated into the insulin resistance syndrome in 1993 [15,16]. Furthermore, HHcy and hyperuricemia could also be added to the “H” phenomenon, particularly since HHcy is an independent risk factor for CCVD, HTN, and vascular stiffening, and a biomarker for impaired FOCM [17]. It is important to note the inclusion of toxic free fatty acids (FFAs) to the left of the hyperlipidemia arm and just above the obesity epidemic (silver box) and present in the earlier simplistic syndrome X (Figure 1). While short-chain, saturated FFA are detrimental and promote accelerated atherosclerosis, it is important to note that the long-chain n-3 polyunsaturated fatty acids are beneficial with antiatherogenic, anti-inflammatory and antithrombotic properties. 

Notably, hyperleptinemia is included and LR (encircled lower mid-Figure 1) and WAT and VAT are important to the development of hyperleptinemia and LR in the MetS. Finally, EC activation—dysfunction and eNOS uncoupling with decreased bioavailable NO are extremely important to the development and progression of the MetS reloaded and its increased risk for the development of CCVD and T2DM (Figure 1) [12,14]. Interestingly, note that the WHO is one of the only organizations that includes microalbuminuria in its definition of the MetS, and that microalbuminuria may reflect not only endothelial activation and dysfunction with decreased NO and increased glomerular capillary permeability, but also endothelial activation systemically [18].

Indeed, the MetS has a rich history along with the various key obesity models that have been created to study the role of obesity. MetS, systemic IR and BIR, T2DM, and leptin—LR along with the discovery of the *ob*/*ob* or the *fa*/*fa* gene and the discovery of leptin in 1994 [19,20]. This important discovery resulted in a paradigm shift in regard to the metabolic, secretory, and endocrine roles played by adipose tissue in addition to a better understanding of the MetS. 

The following timelines of discovery provide the background knowledge that was necessary to develop our understanding and enabled the current concept of the MetS that has been important for obesity and T2DM research that is ongoing to date and its implications with obesity (Figure 3) [19,20,21,22,23,24,25,26,27,28,29,30,31]. 

Alberti et al. (2009), held a joint international task force meeting in an attempt to harmonize the definition and discuss the various risk factors for the diagnosis of the MetS. This task force emphasized the importance of identifying those with the MetS so that they could receive the appropriate treatment by their healthcare providers to improve these risk factors and the increased risk of CCVD (two times higher) and T2DM (five-fold higher) as compared to non-MetS individuals. Furthermore, they made the decision to retain an elevated waist circumference as a risk factor with the realization that the cut points may be different for different ethnic groups, which would be determined later by other task forces [32]. 

Throughout this overview, the use of various preclinical obese, insulin- and leptin-resistant preclinical diabetic rodent models will be utilized in order to provide a better understanding of how the multiple risk factor and variables of the MetS result in cellular and organ ultrastructural remodeling and functional abnormalities. Recently, Panchal and Brown (2011) have discussed the use of rodent models to study the MetS, and concluded that the ideal model to study the parallels between rodents and humans is the diet-induced obesity (DIO) model, where subjects are fed a high fat and high sucrose/glucose diet [33]. Our group has studied multiple obesity models including DIO models; however, most of our transmission electron microscopy (TEM) studies have utilized the *db*/*db* leptin receptor-deficient mouse models with obesity, elevated leptin, LR, IR, and diabetes to identify cellular remodeling changes in multiple organ systems. 

The addition of novel emerging risk factors and variables may allow for a better understanding of the MetS and earlier recognition for individuals and their healthcare providers to prevent the development of CCVD and T2DM. By providing new knowledge and novel insights, it is hoped that further research in this exciting field of study may be stimulated in order to decrease the risk of developing CCVD and T2DM along with their complications in these at-risk individuals. 

The specific aim of this overview is to add novel risk factors, emerging variables, and possible biomarkers to the existing traditional risk factors associated with the MetS. In the following Section 2, Section 3, Section 4 and Section 5, each of the four traditional risk factor arms of the letter X will be discussed in greater detail. Additionally, the more novel non-traditional emerging risk factors and variables are discussed. These include endothelial cell activation/dysfunction, endothelial glycocalyx remodeling (Section 6), and metainflammation including gut microbiota dysbiosis, adipokines, cytokines/chemokines, adipose-derived extracellular exosomes, micro RNAs, and long-non-coding RNAs (Section 7) (Figure 1). 

## 2. MetS Reloaded, Hyperlipidemia—Atherogenic Dyslipidemia

The late Russell Ross was a pioneer and a giant in the understanding, development, and progression of atherosclerosis acting as an inflammatory disease of the arterial vessel wall [34]. Ross felt that this injury to the arterial vessel wall was largely due to elevated levels of cholesterol and particularly LDL-C, modified or oxidized LDL-C, and now we know that the small-dense LDL-C is extremely proatherogenic and proinflammatory. Importantly, the toxic lipids of atherogenic dyslipidemia that are associated with the MetS act as an injury to the intima ECs and the subintimal space. Furthermore, these atherogenic lipids will serve as an injury to the endothelium of the arterial vessel wall with an ensuing response to injury wound healing remodeling, which will result in accelerated atherosclerosis in the MetS and T2DM [12,14]. 

Hyperlipidemia or dyslipidemia in the MetS (Figure 1) is characterized by an atherogenic dyslipidemia, which consists of elevated triglycerides and atherogenic modified LDL-C; oxLDL-C; small dense LDL-C; and decreased atheroprotective HDL-C [35,36]. Additionally, non-HDL-C is now included to demonstrate not only increased VLDL-triglycerides but also remaining remanent atherogenic lipoproteins including lipoprotein (a) (Lp(a)), LDL-C, intermediate-density lipoprotein, and VLDL-C along with elevated toxic, short chain, saturated FFAs, which play an important role in atherogenic dyslipidemia but are often not included in its definition [35]. For example, the elevated triglycerides (triacylglycerols) in atherogenic dyslipidemia associated with the MetS are lowered by long-chain n-3 polyunsaturated fatty acids (long chain n-3 PUFA (LC n-3PUVA) including ω-3 fatty acids) due to a decrease in VLDL via a decreased sterol receptor element-binding protein-1c and decreasing β-Oxidation in mitochondria and peroxisomes [35]. Additionally, LC n-3 PUFAs are known to significantly decrease the risk of fatal coronary heart disease (CHD) [37]. The exact mechanisms through which long-chain n-3 PUFA has an effect on CHD are not completely established, but it is thought that they might include decreased fasting and postprandial triacylglycerol (triglyceride) levels, decreased arrhythmias, modulation of platelet aggregation affecting thrombus formation, and decreased synthesis of pro-inflammatory molecules [37]. The mechanistic relationship between long-chain n-3 PUFA and inflammation has attracted great interest, and in vitro studies have demonstrated that these specific fatty acids decrease endothelial activation and may affect eicosanoid metabolism and result in improved resolution of inflammation [36]. Indeed, there are a multitude of studies that support the positive role of long-chain n-3 PUFA in reducing the risks of ASCVD and CHD that has been extensively reviewed by deRoos et al. [36]. 

Metainflammation is currently recognized as a significant process in the development and progression of not only atherosclerosis but also CCVD and CHD. As put forth by Ross [34], the instigation of inflammation may well provide the link between hyperlipidemia—atherogenic dyslipidemia and atherogenesis, since inflammatory pathways are known to promote thrombosis, the most serious complication of atherosclerosis responsible for both myocardial infarctions and most transient ischemic attacks and strokes. Importantly, we may be facing the dawn of a new era in regard to lipid-lowering, with newer therapies emerging such as monoclonal antibodies, antisense oligonucleotides, small interfering ribonucleic acids (RNAs), and even the possibilities of vaccination that are emerging as novel treatments with LDL-C still being a primary target, non-HDL-C, which includes all of the atherogenic lipoproteins and which includes LDL-C, VLDLs (which transports endogenous triglycerides, phospholipids, cholesterol, and cholesteryl esters) as a secondary target, and the remanent lipoprotein Lp(a)) [37]. The mechanisms of these novel emerging therapies are beyond the scope of this overview; however, for those who have a greater interest, a full discussion can be better appreciated [38]

Elevated triglycerides are an important part of the atherogenic dyslipidemia in the MetS reloaded. Evidence has accumulated that lowering triglycerides and increasing HDL-C levels with specific drug therapy will reduce the risk of CHD independently of statin therapy or LDL-C lowering. At first, this may seem like a bold statement; however, gemfibrozil therapy in the Veterans Administration HDL Intervention Trial (VA-HIT) led to a triglyceride-lowering of 31%, a HDL-C raising of 6%, and no change in LDL-C. Interestingly, US military veterans with known coronary artery disease at the outset of this trial experienced a 24% reduction in coronary events during follow-up [39]. 

The important role of excess omental—visceral white adipose tissue depots along with the excess subcutaneous fat depots, still remain an important tissue for the synthesis of excess leptin production, LR, and chronic sterile metainflammation that is so very important for the development of accelerated atherosclerosis, ASCVD, and CCVD [12,14]. The role of fat is especially important in obese models that are available to study diet-induced obesity (DIO). Western models, due to high sucrose, fructose and fat feeding, the leptin receptor-deficient *db*/*db* models, and the novel leptin-deficient BTBR *ob*/*ob* models are important models to study. Additionally, the early spontaneous mutated obesity Zucker obese *fa*/*fa—ob*/*ob* model discovered by the husband-and-wife team of doctors Zucker LM, Zucker TF in 1958 is a rat model of spontaneous genetic mutation of leptin receptor deficiency [22] that allows for the demonstration of excessive visceral and subcutaneous adipose tissue accumulation (Figure 4) [40].

## 3. MetS Reloaded, Insulin, Hyperinsulinemia, IR, Leptin, Hyperleptinemia, and LR

Insulin and adipose tissue-derived leptin hormones from both white adipose and visceral adipose tissue act centrally and are important in regulating food intake and adiposity in rodent models and humans [30,41]. Additionally, IR and compensatory hyperinsulinemia are associated with multiple abnormalities associated with the MetS. These abnormalities include impaired insulin signaling of the phosphoinositide 3-kinase/protein kinase B (PI3Kinase/AKT) pathway that is associated with serine hyperphosphorylation of the insulin receptor substrate 2 (IRS2) signaling pathway, and decreased insulin-stimulated NO production with decreased vasodilation and proconstrictive, prothrombotic, and pro-atherosclerotic effects on the arterial vessels with accelerated atherosclerosis [12,14]. In addition, there is an associated increased redox stress, endoplasmic reticulum stress, and promotion of metainflammation as previously discussed and depicted in Section 1. (Figure 2). 

LR and hyperleptinemia are known to be present in human obesity [30,42,43,44,45]. Central and peripheral leptin resistance may be present under pathophysiological conditions such as hyperlipidemia, hypertension, atherosclerosis, obesity, inflammation, and IR of the MetS [46,47]. The adipocyte-derived peptide adipokine/hormone leptin in humans has been linked to adiposity and insulin resistance [30,42,43,44]. Even after adjustment for the percentage of body fat in middle-aged men and women, the rate of insulin-mediated glucose uptake has been found to significantly associate with leptin levels [45]. Recent research suggests that leptin may be an important factor linking obesity, the MetS, and CCVD [46]. Further, it is known that leptin may be an important factor in regulating blood pressure and volume under physiologic conditions. However, during conditions of leptin resistance and hyperleptinemia as occurs in obesity of the MetS, this endocrine hormone and adipokine may function pathologically in regards to the development of HTN and CCVD [47]. Thus, leptin and LR seem to be emerging novel players that are playing a central role in the MetS, in that LR is not only associated with obesity but also contributes to obesity and independently affects IR [42,43,44,45,47]. 

### 3.1. MetS Reloaded, Compensatory Hyperinsulinemia, Insulin Resistance (IR), and Compensatory Hyperamylinemia 

As previously mentioned, the importance of IR from an epidemiologic standpoint is that IR affects nearly one-third of the US population and is known to precede the development of T2DM [12,14,30]. Also, it has been recently estimated that the MetS prevalence was more than one-third in the US population for all sociodemographic groups, and these numbers increased from 1988 on [30,48]. 

Hyperinsulinemia and hyperamylinemia are the result of islet β-cells’ compensatory response to overcome cellular insulin resistance to glucose uptake in the MetS. MetS affects approximately 47 million or more Americans [49,50]. It is thought that approximately 20% will develop overt T2DM, while the remaining 80% would be capable of developing compensatory hyperinsulinemia and hyperamylinemia. This compensation for the underlying IR is due to the increased β-Cell secretion of both insulin and amylin at least for a period of time through the processes of pancreatic β-cell expansion, hypertrophy, increased β-cell endoplasmic reticulum (ER) stress, and hyperplasia. However, the compensatory hyperinsulinemia (37.6 million = 80% of 47million) that develops in the remaining 80% to prevent the development of T2DM does not come without a ‘price to pay’, in that this chronic compensatory hyperinsulinemia places these patients at risk for HTN, accelerated atherosclerosis—atherogenesis, and subsequent CCVD due to atherosclerotic cerebral vascular and coronary heart disease (CCVD) (Figure 1 and Figure 2) [5,12,14,51].

Because of this increased compensation by producing more insulin and amylin, there would also be an increased risk of islet amyloid deposition due to the amyloidogenic hyperamylinemia and hyperinsulinemia, which are both common to IR of the MetS [52,53]. Incidentally, there was an increased risk for those older obese individuals who developed T2DM during the recent COVID-19 pandemic who also experienced endothelial cell activation and dysfunction [52]. Over time, β-cell failure will develop due to β-cell apoptosis, and T2DM will develop as a result of gluco-lipotoxicity and islet ER and redox stress (Figure 5) [52].

### 3.2. MetS Reloaded, Insulin Resistance, Compensatory Hyperamylinemia, and Islet Amyloid Deposition

Amylin (IAPP) is a 37 amino acid polypeptide synthesized by the pancreatic islet β-cells. It is co-synthesized and co-packaged within the insulin secretory granule (ISG) by the endoplasmic reticulum and is subsequently secreted into the systemic circulation. Excessive amounts of amylin due to insulin resistance may also be deposited within pancreatic islets as misfolded proteins termed islet amyloid polypeptide (IAPP) [[52],[53],[54]). This excessive deposition of IAPP within the islets is known to be associated with and contribute to the development of T2DM (Figure 5E) [52,53,54,55,56,57,58]. Additionally, Cooper et al., have found that excessive amylin (hyperamylinemia) was capable of stimulating renin and the renin-angiotensin-aldosterone system (RAAS) (Figure 1) [59]. Indeed, islet amyloid polypeptide (hIAPP)—islet amyloid is a definite hallmark remodeling change found within pancreatic islets and is known to be present in a substantial portion of human individuals with T2DM [52,55,56,57,58,60,61,62]. Additionally, Despa and colleagues have recently been able to demonstrate that islet amyloid is also capable of depositing within the CNS to form an amyloid niche. This amyloid niche is thought to instigate and contribute to increased amyloid beta deposition and progression in the brain and possibly accelerate cognitive decline in late-onset Alzheimer’s disease and neurodegeneration [63,64,65,66,67]. In the past two decades, there has been a better understanding of the role of hyperamylinemia and pancreatic islet amyloid in T2DM since its presence was initially described by Eugene Opie in 1900–1901 [68,69].

Mitochondria (Mt) are vital to the normal function of pancreatic islet beta cells since the synthesis and secretion of the islet insulin secretory granules require a constant supply of energy in the form of adenosine triphosphate (ATP) [11]. Islet glucose and lipids serve as energy nutrients to the islet beta cells, which sense rising glucose to activate insulin secretion. As the surrounding systemic glucose levels rise within the islet, there will be an increased activity of the mitochondrial electron transport chain (ETC) to produce ATP, which results in the following sequence of events: The rise in the ATP/ADP ratio is responsible for the closing of the ATP-sensitive potassium (K+) channel, which results in the depolarization of the islet beta-cell plasma membrane and serves as a signal to open the voltage-gated calcium channels, and this, in turn, is responsible for the exocytosis of insulin secretory granules and insulin [11]. If there is Mt dysfunction with the decreased production of ATP, there will be less ATP generated and this will interfere with the proper secretion of insulin secretory granules and the dysfunction of insulin secretion. Additionally, if the beta-cell mitochondria are further damaged due to increased RONSS and the RSI, then the abnormally remodeled aberrant Mt (aMt) with the loss of Mt matrix electron density and crista become leaky, allowing cytochrome c to be released with the activation of beta-cell caspases, which will result in beta-cell loss due to apoptosis (Figure 5E and Figure 6) [11,52].

In addition, when both aMt and ER stress are found to co-occur within the pancreatic islet beta-cells, there will be apoptotic signals from ER stress and the unfolded protein response (UPR) and cytochrome c leakage from aMt that signals caspases for the beta-cell to undergo apoptosis [52,70,71]. Anello et al., were able to demonstrate that human islet beta-cells showed an impaired insulin secretory response to glucose and an associated marked alteration of Mt function and morphology that was associated with an increased expression of uncoupling protein 2 (UCP2). UCP2 was thought to be due to increased glucose overload that was associated with a lower ATP, a decreased ATP/ADP ratio, and a consequent reduction of insulin release [71]. Also, they found that mitochondria numbers were similar between the control and T2DM in beta cells; however, the volume of the Mt were statically increased and hyperlucent with the loss of the electron-dense mitochondria matrix and the loss of cristae in shown in Figure 5 [71]. Importantly, the accumulation of aMt and impaired mitophagy in islet beta-cells are associated with impaired synthesis of insulin secretory granules (the paucity of ISGs, as shown in Figure 5) and deficient ATP production and thus a decrease in insulin secretion and the eventual loss of beta cells via apoptosis (Figure 5F).

Furthermore, it has been recently proposed that there exists a critical interdependence between Mt dysfunction and T2DM that is found in the pancreatic islet β cells [72,73]. Indeed, T2DM is a multifactorial and polygenic disease which is associated with mitochondrial dysfunction due to aMt remodeling phenotypes. This central defect of aMt contributes to both insulin resistance and β-cell dysfunction and eventually β-cell failure via apoptosis in T2DM. Therefore, it is important to note that Mt dysfunction (aMt) and T2DM may be bidirectional (Figure 7).

### 3.3. MetS Reloaded, Leptin, Hyperleptinemia, and LR

The discovery of leptin in 1994 provided a paradigm shift in how the adipose tissue is currently viewed (Figure 3A) [19,20], such that leptin is thought to be a predictive marker of the MetS in humans [74,75]. Furthermore, leptin and adiponectin are two of the most abundant adipokines secreted from adipocytes that have been implicated in the development of T2DM and CCVD and are associated with MetS, VAT, IR, β-cell dysfunction, inflammation, arterial stiffness, and subclinical atherosclerosis, at least in children [76]. In chronic obesity, it is commonly accepted that leptin becomes elevated, while adiponectin becomes decreased and results in a decreased adiponectin/leptin ratio, which predisposes to accelerated atherosclerosis and increased risk of CCVD [77]. Additionally, hyperleptinemia and hypoadiponectinemia are currently thought to be possible therapeutic targets of the MetS (Figure 8) [78,79].

Notably, Leptin is also an emerging trigger for stroke [80,81,82], and associates with obstructive sleep apnea [83].

The MetS increases with aging, and the global society currently has one of its oldest populations in our history [43,44,82,83,84]. Even though resistance to leptin is known to occur in aging [85,86], one must keep in mind that Leptin and Lr still remain incompletely understood, and we must continue to closely monitor future research on leptin and how it applies to human clinical medicine. For example, it has recently been shared that leptin resistance during aging may be independent of fat mass in preclinical rodent modes [87]. Thus, in relation to the MetS, we must be cautious to ensure that we examine not only for age but also for gender differences [87]. Since leptin and adiponectin are the two most abundantly produced adipokines secreted from adipocytes that play an important role in glucose homeostasis, inflammation, and accelerated atherosclerosis in chronic obesity, it is important to keep a close watch on further developments in this field of study [85,86].

## 4. MetS Reloaded, Essential Hypertension (HTN), and Vascular Stiffening

HTN may be defined as the elevation of blood pressure of unknown cause (in contrast to secondary hypertension with a known cause), and it frequently clusters with other cardiovascular risk factors such as obesity, IR, LR, diabetes, hyperlipidemia, and aging in the MetS reloaded [88].

The MetS reloaded is associated with HTN and is one of the core features along with obesity, hyperlipidemia, hyperinsulinemia, IR, LR, and hyperglycemia (Figure 1). HTN is strongly associated with hyperinsulinemia, IR and LR (Figure 9) [89,90,91,92,93].

Leptin’s effect on HTN is mainly through its renal, sympathetic nervous system, and vascular functions [94]. The MetS reloaded is very complicated and consists of a complex network of interconnected pathways with regard to the development of HTN. While some of these pathways still remain incompletely understood, the following figure provides further insight as to how this complex network is interconnected with the various components of the MetS reloaded (Figure 10) [88,89,90,91,92,93,94,95,96,97].

In regards to vascular-arterial stiffening, Lopes-Vincente et al., determined that vascular or arterial stiffening as measured by pulse wave velocity in the carotid-femoral segments was already present in a cohort of individuals with newly diagnosed MetS as compared to those without MetS. Furthermore, they found that aging and the number of risk factors (greater than the three required to diagnose MetS) were associated with increased vascular stiffness [97]. In addition, Schillaci et al. were able to show that the MetS influences arterial functional properties that are related to cardiovascular risk, and provide a pathophysiological framework for understanding the associations between MetS and vascular stiffness related to cardiovascular morbidity and mortality [98]. These findings are not surprising since the pathophysiology for the development of HTN in the MetS was associated with so many interacting pathways (Figure 9 and Figure 10) involved with alterations in vascular functions and the known association with vascular remodeling as in atherosclerosis discussed in Section 2. Importantly, the Framingham Heart Study has confirmed that vascular stiffness is an independent predictor of cardiovascular morbidity and mortality in the general population, hypertensive individuals, and the elderly [99]. Additionally, vascular stiffness is known to be a consequence of pathophysiological alterations, which include vascular smooth muscle cells (VSMCs), ECs, elastin, and extracellular matrix (ECM) (Figure 9 and Figure 10) [99,100,101]. Additionally, Yan et al. have proposed that HTN in the MetS is a form of salt-sensitive HTN that is largely based on the dysfunctional Na/K-ATPase enzyme that is redox sensitive [102]. Furthermore, there is known to be a lack of IR in the kidney, and there is increased renal sodium reabsorption due to hyperinsulinemia in the MetS that leads to sodium retention and the resultant salt-sensitive HTN [103,104].

Notably, obesity and the hypothalamic-pituitary-adrenal (HPA) axis dysfunction are implicated in RAAS activation in the MetS reloaded. In Figure 1, note the yellow line that extends downward from the boxed-in HPA axis dysfunction to the arrow indicating RAAS activation of Ang II and aldosterone and the dashed arrow from RAAS to the HPA axis, which supports a bidirectional activation of the HPA axis and the RAAS. In addition, this is responsible for the sympathetic nervous system activation. Previously, our group found considerable remodeling within the adrenal gland zona glomerulosa cells of the diet-induced obesity Western models at 12 weeks. The remodeling changes consisted of zona glomerulosa hypertrophy, hyperplasia, capillary dilation, and endoplasmic reticulum stress [8].

## 5. MetS Reloaded, Hyperglycemia, Mitochondrial Dysfunction, and Importance of the Reactive Oxygen Species Interactome (RSI)

There is a very strong overlap between IFG and IGT in the MetS reloaded, and this points to the important role of hyperglycemia in the upper left-hand side of the MetS reloaded (Figure 1) [105]. T2DM is a multifactorial polygenic disease that may be characterized as a chronic metabolic–endocrine disorder. T2DM is associated with IR or insulin deficiency, which results in hyperglycemia, IFG-IGT, and T2DM. This glucotoxicity promotes oxidative stress and chronic inflammation [105]. Furthermore, T2DM is a risk factor for macrovascular (accelerated atherosclerosis and vascular stiffness) [14] and microvascular end-organ pathologies such as retinopathy [106], neuropathy [107], nephropathy [108], vasculopathy-intimopathy [12], isletopathy [109], accelerated atherosclerosis (atheroscleropathy) [14], cardiomyopathy [110], and diabetic encephalopathy-cognopathy (cognitive dysfunction), which associates with an increased risk for age-related neurodegenerative diseases such late-onset Alzheimer’s disease (LOAD) [8]. Also, macrovascular and microvascular diseases are at the crossroads of obesity, IR, LR, T2DM, and HTN [111].

IFG-IGT and T2DM result in hyperglycemia, which is associated with increased ROS production via the following six pathways that increase ROS: (1) glucose autoxidation; (2) polyol flux; (3) hexosamine flux; (4) advanced glycation end product(s) (AGE); (5) increased receptor for AGEs (RAGE) and AGE/RAGE interactions; and (6) increased protein kinase C and mitochondrial overproduction of ROS in macrovascular endothelial cells by the increasing FFA flux and oxidation [110,111,112]. Additionally, the MetS reloaded is associated with leaky aMt, which leak superoxide and contributes to the overall ROS, which interacts with RSI consisting of RONSS to result in excessive oxidative stress. This cumulative increase in oxidative stress is responsible for the inactivation of two critical antiatherosclerotic enzymes eNOS and prostacyclin synthase and defective angiogenesis in response to ischemia. This oxidative stress also activates a number of proinflammatory pathways, which enables ROS to induce inflammation via increased nuclear factor-kappa B (NF-*κB)* and downstream cytokines/chemokines, and inflammation to induce ROS, which creates a vicious cycle. These pathways are also responsible for long-lasting epigenetic changes that continue to drive the persistent expression of proinflammatory genes even after glycemia is normalized, and are thus important in creating hyperglycemic memory [111,112]. There are several lines of evidence which indicate that all of the six previous listed causes for increased ROS in hyperglycemia are activated by a single important upstream event, and that is the mitochondrial excess production of mitochondria ROS (mtROS) superoxide (^•^O_2_^−^) [110,111,112,113]. Importantly, aMt leak superoxide or mitochondria ROS (mtROS), which induce deoxyribonucleic acid (DNA) strand breaks and activate nuclear poly (ADP-ribose) polymerase, resulting in decreased glyceraldehyde 3-phosphate dehydrogenase (GAPDH), and subsequently increase the previous causes for increased oxidative stress in the MetS reloaded (pathways1-6) [11,100,111,112,113].

Over the past 17 years, the identification of aMt have been a constant, recurrent, and unifying finding utilizing TEM in multiple organs including the endothelium [11,113,114,115,116] identified in our obese, MetS, IR, LR, IGT, and diabetic preclinical rodent models with hyperglycemia. Other organs with aMt included the kidney [11,40,116,117,118], the skeletal muscle [11,40,118], the cardiac muscle [11,40,94,119,120], the liver [94], the aorta [11,121], VSMCs [11,40], the pancreatic islets [40,52,53,54] and the brain [6,8,11,30,114,115,116,122]. Two recent topical reviews discuss the role of Mt as metabolic hubs and unifying organelles in multiple organs of obesity; IR, MetS, and T2DM are available for those who wish to view additional ultrastructural images of normal Mt and aMt [11] and the role of the Mt in the MetS [72].

Brownlee et al., have previously noted that the upstream aMt and the increased mtROS may be extremely important due to the additive damage from hyperglycemia-induced ROS [112,113]. This is especially important when there are excessive mtROS due to leaky aMt. Indeed, the remodeling to an aMt leaky phenotype is a common finding in multiple organ systems of obesity, IR, LR, MetS reloaded, and T2DM. aMt contribute to the overall increased oxidative stress, and its mtROS will interact with RONSS and the RSI to further increase the overall oxidative stress in the MetS and T2DM. Since FOCM is known to be compartmentalized to the mitochondria, cytosol, and nucleus, it is important to next examine the role of impaired FOCM in the MetS reloaded [17].

### 5.1. MetS Reloaded and Impaired Folate-Mediated One-Carbon Metabolism (FOCM) in the Metabolic Syndrome Reloaded

FOCM is a complicated metabolic network of interdependent biosynthetic pathways and cycles that are known to be compartmentalized in the cytoplasm, mitochondria, and nucleus (Figure 11) [17].

Methionine, tetrahydrofolate (THF), and vitamin B12 are primarily supplied by dietary sources in order to supply the necessary substrates for methionine and folate cycles. Therefore, B12 and methionine are placed in a central position of the methionine and folate cycles (Figure 11). In addition, note that vitamin B12 (cobalamin) is essential for the demethylation of HCY to methionine [17,123]. If B12 and/or methionine synthase (MS) are deficient, there will be elevations of the clinical biomarkers of impaired FOCM such as HHCY and methylmalonic acid. Additionally, vitamin B12 is also important in the production of succinyl Coenzyme A that is necessary for the proper function of the TCA/Krebs cycle. Folic acid (folate or vitamin B9) is important for the FOCM along with B12 to properly supply Succinyl-CoA, while glycolysis provides oxaloacetate from pyruvate to the tricarboxylic acid TCA cycle to provide NADH and FADH2 to the ETC to produce ATP by the mitochondria [17,123]. Furthermore, note that when there is an intact FOCM metabolic state, Hcy can be either demethylated to methionine via the methionine synthase with its essential cofactor vitamin B12, or enter the cystathionine beta synthase pathway via an intact vitamin B6 pathway [17]. The ETC generates ATP via complexes I–V to generate the energy currency of ATP. Importantly, if there are excess nutrients in the form of glucose, sucrose, fructose, and fat delivered to the ETC at complex I, there will be a generation of excessive ROS in the form of superoxide (^•^O_2_^−^ ) or mtROS. An excess of mtROS will then interact with extra-mitochondrial or cytosolic nitrogen and sulfur to form RONSS, which induces the RSI, wherein ROS-induced ROS release (RIRR) comes into play and will accelerate RONSS RSI and redox stress. Importantly, note that if the methyltetrahydrofolate reductase (MTHFR) enzyme (encircled with a red-dashed line) becomes dysfunctional, HCY in the methionine cycle will also become elevated, as depicted by the red-dashed line from MTHF to HCY (Figure 11A) [17].

In addition, formate plays an important role in providing proper nucleus function and maintenance of its cellular structure in homeostatic conditions. Serine, glycine, methionine, and choline are necessary substrates to provide formate in order to provide proper mitochondrial function to the nucleus to produce purines, thymidylate and methionine to fulfil their role in the nucleus. Formate is primarily produced within the mitochondria, and is subsequently secreted and enters the nucleus via nuclear pores to provide FOCM to the nucleus for proper chromatin, histone modeling, and normal function [17]. Notably, deoxythymidine monophosphate is synthesized in the cytosol, nucleus, and mitochondria, whereas purine synthesis and methionine synthesis take place within the cytosol. Mitochondrial FOCM generates formate for cytosolic and nuclear FOCM and biosynthetic precursors for mtDNA synthesis and mitochondrial protein translation. Thymidylate synthase converts deoxyuridine monophosphate to deoxythymidine monophosphate (dTMP) in a 5,10-methylene-THF-dependent reaction. It is important to note that mitochondrial SAM (Mt SAM) is derived from cytosolic SAM. Additionally, the Krebs cycle also resides within the mitochondria and provides reduced nicotinamide adenine dinucleotide and the reduced flavin adenine dinucleotide to the ETC for ATP production [17].

Folate (folic acid, essential vitamin B9) belongs to a family of enzyme cofactors that carry chemically activated 1-carbon units to formate, formaldehyde, and methanol. Folate 1-carbons are necessary for the synthesis of purine, thymidylate, and the remethylation of HCY to methionine. Furthermore, methionine is an essential amino acid that is utilized in protein synthesis that can also be adenosylated to S-adenosylmethioinine (SAM), which is responsible for the methylation of protein cytosine bases of DNA (important for nuclear chromatin structure), neurotransmitters, phospholipids, and other protein molecules (Figure 11) [17,123].

FOCM is a metabolic network of interdependent biosynthetic pathways and is important for physiologic maintenance and proper cellular homeostasis due to its role in purine and thymidylate synthesis. FOCM is compartmentalized in the nucleus, mitochondria, and cytosol (Figure 11B) [17]. However, it becomes impaired in the hyperglycemia (IFG, IGT, and T2DM) of the MetS reloaded [17,123,124,125,126,127]. Additionally, Hcy is known to be elevated in T2DM and is associated with HHCY. Importantly, HHCY may now be considered a clinical biomarker for impaired FOCM [17,123,125,126,127,128]. HHCY would also add to the existing six factors for increased ROS production due to hyperglycemia and the aMt [11,109,112,113]. Hyperglycemia-induced ROS could interact with not only the aMt (mt ROS) but also the HHCY-induced ROS, which could subsequently interact with the RONSS of the RSI, and these multiplicative ROSs could form a situation in which ROS becomes self-perpetuating, which is a condition of ROS-induced ROS release, creating a vicious cycle of excessive oxidative stress to cells and tissues in IFG, IGT, and T2DM [17,129]. The folate and methionine cycles and the transsulfuration pathway are all extremely important not only in the mitochondria but also in the cytosol and nucleus [17]. In order for the normal functioning of the FOCM, there must be an adequate supply of water-soluble B vitamins including B12, B9 (folic acid or folate), and B6 (pyridoxal 5′-phosphate) as they relate to FOCM (Figure 1 and Figure 11).

Interestingly, an intact properly functioning FOCM metabolic pathway with adequate amounts of B vitamins is now being considered to play the role of providing antioxidant roles at least in stroke [130]. It is known that folate (B9) is capable of aiding in the role of preventing the complete BH4 oxidant stress to this cofactor that is necessary for the production of endothelial-derived NO. Additionally, the FOCM metabolic pathway is necessary to convert the potent oxidant of HYC—HHCY via the cystathionine beta-synthase enzyme (CBS) with adequate amounts of vitamin B6 and serine to generate glutathione via the conversion of cystathionine to cysteine to GSH, a potent cellular derived intracellular antioxidant (Figure 11). In addition, cysteine may enter the transsulfuration pathway. Indeed, the FOCM pathways are far-reaching and play multiple important roles in providing proper cellular homeostasis [17]. FOCM is important to metabolize HCY (a potent oxidant) to methionine, which when elevated is a known risk for developing conditions such as stroke and cellular redox stress. In preclinical animal models and human individuals, it is possible to lower elevated Hcy with vitamin B12 and folate; however, the utilization of nutrient therapies such as B vitamins to improve the ischemia associated with acute stroke has yet to be convincingly demonstrated [130].

Impaired FOCM and HHCY are associated with multiple common clinical diseases, which include developmental anomalies with neural tube defects; CCVDs including stroke and cognitive decline, and intestinal malignancies (Figure 12) [17,130].

Mild to moderate HHcy occurs in approximately 5–7% of the general population [131]. There are known to be genetic mutations in human individuals involving the methyltetrahydrofolate (MTHFR) gene, which result in the impairment of its function and thus allow HCY to accumulate and result in HHCY. This is especially true if there are deficiencies of the essential B vitamins (B9 -folate, B12, and B6) (Figure 11). The most common human genetic variant of the MTHFR gene to date is the primary homozygous C677T (T677T) and the secondary compound heterozygous A1298C + C677T. This genetic variant occurs in 10-15% of the general population and is likely to occur in similar percentages in the MetS population [132,133,134].

## 6. MetS Reloaded and Endothelial Activation (EC*act*) and Dysfunction (EC*dys*)

The endothelium is formed by a thin protective monolayer of endothelial cells and is known to play an important, complex role in vascular biology. Endothelial cell(s) (ECs) are key to vascular hemostasis, tone, leukocyte recruitment, hormone trafficking, and fluid movement from the blood to the interstitial space [135]. The increased endothelial expression of cell-surface adhesion molecules, such as VCAM-1, ICAM-1, and E-selectin, may be defined as endothelial cell activation (EC*act*). Similarly, endothelial cell dysfunction (EC*dys*) may be defined as the decreased synthesis, release, and/or activity of endothelium-derived NO, which results in decreased bioavailable NO [135,136,137]. Notably, EC*act* and EC*dys* are tightly related and share a bidirectional relationship, which results in accelerated atherosclerosis and leads to CCVD by inducing a proconstrictive milieu, vascular smooth muscle cell proliferation, and a proinflammatory state with leukocyte adhesion, platelet aggregation, lipid oxidation, and matrix metalloproteinase (MMP) activation [135]. EC*act* may be induced by (1) proinflammatory cytokines, such as TNF-α and IL-6; (2) turbulent blood flow, such as that which occurs at bifurcations and branch points of arteries; (3) AGEs, which are elevated in hyperglycemia and aging, and (4) inflammatory stressors such as metainflammation and plasma membranes’ peptides of gram-negative bacteria such as LPS. These four functions are each important mediators of EC*act* via the activation of the nuclear transcription factor of EC NF-κB. The following TEM images are examples of brain EC (BEC) activation (Figure 13) [6,8,11,17,136].

Over the past decade, the author has been able to identify multiple ultrastructural transmission electron microscopic (TEM) remodeling changes associated with ECact/dys (Figure 14) [6,8,11,17,122,136,137,138].

EC*dys* is defined as the decreased synthesis and activity of endothelium-derived nitric oxide (NO), which is frequently referred to as decreased NO bioavailability [135]. We now know that ECact and ECdys are closely linked in a response to molecular and structural injury to the arterial vessel wall such that a decrease in the bioavailability or loss of NO can lead to ECact, and in a similar manner, ECact can cause ECdys and, thus, the close linkage between these two processes [11,135,136]. Additionally, we now know that NO has a multitude of positive effects on the vascular wall, which includes its anti-inflammatory, antithrombotic, anti-atherosclerotic, and vasodilation properties [125,133,135,136]. This places endothelium-derived NO at the very central core for maintaining vascular homeostasis. Notably, when bioavailable NO is decreased, vascular homeostasis will be disrupted, since ECact and ECdys are closely linked [11,136]. Obesity—hyperlipidemia, hyperinsulinemia—IR, HTN, and hyperglycemia are known to be associated with the decreased bioavailability of NO (EC*dys*), either directly or indirectly, that may result in EC*act* [11,12,136,139]. IR results in the decreased signaling of the eNOS enzyme and its activation and, therefore, the combination of IR and excessive nutrient supply via excess glucose, sucrose, or fructose will increase mtROS via the electron transport chain to result in EC*act* and EC*dys*. Similarly, in manifest T2DM there will be an associated increase in aMt that will leak mtROS—superoxide, and there will be excessive oxidation of the eNOS enzyme cofactor tetrahydrobiopterin (BH4) that ultimately uncouples the eNOS production of NO (eNOS enzyme uncoupling) that will decrease endothelial-derived NO bioavailability [11,12,136,140]. The endothelial constitutive and rate-limiting enzyme (eNOS) is responsible for the conversion of L-arginine to NO and L-citrulline, and eNOS requires the essential cofactor tetrahydrobiopterin (BH4) [12,14,136,140].

Reduced BH4 and/or the substrate L-arginine result in the uncoupling of the eNOS enzyme, and the increased production of superoxide and ECdys with decreased bioavailable NO. Causes for this uncoupling include increased superoxide and peroxynitrite (ONOO-), glucotoxicity, small dense LDL-C, oxidized LDL-C, HHCY, increased mtROS, decreased L-arginine, increased asymmetric dimethylarginine (ADMA), and highly sensitive C reactive protein, plus others that promote oxidative stress to the endothelium [12,14]. In physiologic homeostasis, the endothelium is a net producer of NO; however, in obesity, IR, and especially manifest T2DM, the endothelium may become a net producer of superoxide unless the blood glucose is optimally controlled. When the superoxide is excessive, as in T2DM, a decrease in the ratio of NO/ROS develops [140,141,142,143]. Indeed, reduced endothelial NOS enzyme activity and eNOS uncoupling result in decreased NO bioavailability, and may be considered as the main factor underlying endothelial dysfunction that occurs in the MetS reloaded (Figure 15) [14,136,141,142,143].

In T2DM there is an increase in aMt, which suggests that the excess glucose may be responsible for the phenotypic ultrastructure of the aMt and causes the increased production of mtROS and superoxide. In addition, the remodeled aMt that are frequently found in T2DM are leaky and allow an even greater release of mtROS (Figure 6). This increase in mtROS will in turn produce eNOS uncoupling and decreased NO bioavailability to vascular tissues [144,145]. IR and LR are central to the MetS reloaded and are coupled to the function of the Mt and the Mt dysfunction is also coupled to IR and LR. This complex interaction is reflected in a finding from Peterson et al., as they found that IR in the elderly population was associated with decreased mitochondrial ATP production, a reduction of Mt RNA, and the decreased activity of oxidative phosphorylation complexes [146]. The association of IR, LR, MetS, and T2DM are related to the function of the Mt; however, it is still not completely known as to whether Mt dysfunction results from or causes IR [11,13,14,144,145,146,147,148,149,150]. In our studies in obese, insulin-resistant, and diabetic models, we have found the following abnormalities in Mt: impaired Mt biogenesis, Mt fragmentation, increased fission, and impaired mitophagy with the accumulation of aMt (Figure 6). The constant and persistent finding in these models of aMt seemed to be a unifying theme of our ultrastructure findings, which strongly suggested impaired mitophagy. Indeed, Mt dysfunction is implicated in the development of IR, LR, and T2DM, and plays a very significant role in the development of diabetic complications.

MetS Reloaded and the Endothelial Glycocalyx

The endothelial glycocalyx (ecGCx) is often referred to as the endothelial surface coating layer, and consists of a unique extracellular matrix (ECM) that acts as an initial barrier between the vascular wall and its luminal contents. Lanthanum nitrate perfusion fixation staining of the ecGCx has recently been found in our laboratory to be both reliable and reproducible, and others who are utilizing this technique have had similar results (Figure 14) [30,151,152,153,154].

The ecGCx is vasculoprotective and acts as the first barrier of a tripartite BBB, including (*i*) ecGCx; (*ii*) brain endothelial cell(s) (BEC) and its basement membrane (BM) of the NVU; and (*iii)* BEC BM and astrocyte foot processes or end feet (ACfps) of the extravascular compartment [30,155] of the NVU. The neurovascular unit (NVU) consists of (1) ecGCx; (2) BEC; and (3) abluminal BM, pericytes (Pcs) and pericyte foot processes (Pcfps) within the BM, and ACfps in addition to the paracellular tight and adherens junctions (TJ/AJs) between each BEC consisting of occludins, claudins, and junctional adhesion molecules in addition to vascular endothelial cadherins (VE-Cadherins) [17,30,136,156]. The BECs are very specialized, and their ecGCx are thicker and more continuous when compared to the heart and lung capillary ECs [152]. The BECs are responsible for the synthesis of not only the luminal ecGCx (with contributions from plasma albumin, orosomucoids, fibrinogen, glycolipids, and glycoproteins), but also the synthesis of the abluminal ECM of the BM [17,30,136,156,157]. Additionally, BECs are primarily responsible for the synthesis of the TJ/AJ, JAMs, and VE-cadherins that form the paracellular barrier of the BBB. The ecGCx is primarily synthesized by BECs, with some contributions by plasma fibrinogen, orosomucoids, albumin, glycoproteins, and glycolipids [11,17,30,136,156,157]. The ecGCx is anchored to the BEC luminal plasma membranes by highly sulfated proteoglycans (syndecans and glycipans), glycoproteins (including selectins such as various cellular adhesion molecules and integrins), and non-sulfated hyaluronan (a glycosaminoglycan) via a BEC cluster of differentiation 44 (CD44). Hyaluronan may also be free-floating (unbound) or attached to the assembly proteins such as the BEC hyaluronan synthases, or form hyaluronon-hyaluronon stable complexes. Furthermore, the ecGCx is also anchored to the caveolae via the proteoglycan (glypican), and this plays a key role in the mechanotransduction of the BEC luminal fluid shear stress-induced synthesis of essential endothelial cell-derived NO via glypican caveolae interactions located within the BEC lipid rafts. The net negative charge of the ecGCx is largely due to the sulfation of glycosaminoglycan side chains, which allow for strong electrostatic binding to the polyvalent cation lanthanum nitrate (La(3+) nitrate) (LAN). This feature of the ecGCx allows it to bind strongly to lanthanum nitrate for its identification by TEM (Figure 14) [17,136,153,154,157].

Impaired FOCM with HHCY, oxidative/redox stress, and metainflammation can be damaging to the ecGCx and contribute to endothelial cell activation and dysfunction with detrimental effects on the vascular tissue that predispose it to increased vascular inflammation and a prothrombotic state and ischemia, which is also an inducer of ecGCx loss. Additionally, aberrant mitochondria (aMt) within the vascular endothelial cells may also be associated with the attenuation and/or loss of the ecGCx along with systemic aMt that are associated with increased oxidative stress and the RONSS interactome, impaired FOCM, and HHCY in the MetS reloaded (discussed in previous Section 5.1 and Section 6). [11,12,17,124,136].

Leptin is important for the appropriate maintenance of the ecGCx. For example, when leptin is deficient as in the novel genetically leptin-deficient, obese, diabetic BTBR *ob*/*ob* preclinical model (due to a leptin-deficiency mutation with no measurable leptin [31]) the ecGCx was attenuated and/or lost. This aberrant ecGCx remodeling was observed in both the frontal cortical layer III and hippocampus regions. However, following leptin administration, the ecGCx appeared to be restored (Figure 16) [30].

Thus, the absence of leptin results in the attenuation and even loss of the ecGCx, which may be an early change in BTBR *ob*/*ob* models [30]. It is also known that adiponectin is important in maintaining the function of the EC, and that concurrent hyperlipidemia is associated with decreased adiponectin, which is known to be present in BTBR *ob*/*ob* models [158]. Thus, a decrease in adiponectin in BTBR *ob*/*ob* models may also be playing an additional role in ecGCx remodeling.

When the NVU ecGCx becomes dysfunctional, attenuated and/or lost via shedding, as in the obese, insulin-resistant BTBR *ob*/*ob* model there will be a decrease in bioavailable NO to signal pericytes in the capillary NVU as well as increased neurovascular unit permeability [11,17,136]. Additionally, we have previously been able to demonstrate ACfp retraction in the obese, insulin-resistant, leptin-resistant diabetic *db*/*db* model [8,114]. Each of these remodeling changes could result in a regional loss of communication between neurons and the neurovascular unit (NVU), and result in the loss of neurovascular coupling. The attenuation and/or loss of the ecGCx could additionally result in the loss of regional mechanotransduction and result in endothelial dysfunction and possibly the loss of neurovascular coupling, which would result in decreased regional cerebral blood flow and ischemia [159,160,161].

The intact ecGCx and Mt of brain ECs are of great importance in maintaining the NVU blood-brain barrier (BBB) homeostasis. Since ecGCx loss appears to be an early event in the BTBR *ob*/*ob* models, this could add to the oxidative stress in brain ECs and contribute to an increase in NADPH oxidase-derived ROS. The attenuation and/or loss of the ecGCx in T2DM can certainly have multiple pathophysiological outcomes, which include increased vascular permeability, edema formation, increased adhesion of circulating inflammatory cells, accelerated inflammatory processes, the activation of the coagulation cascade, and platelet aggregation [11,17,150,151,152,153,154,155,156,159,160,161]. Furthermore, excess nutrients such as increased glucose and fatty acids that occur in obesity, MetS, IR, LR, and T2DM will result in leaky aMt phenotypes and an increase in mtROS, as previously observed in the brains of the obese, insulin-resistant diabetic *db*/*db* models in addition to the increase in NADPH oxidase [43,45,46,47]. Indeed, there seems to be a bidirectional relationship between aMt, MtROS, and attenuation and/or loss of the BEC ecGCx (Figure 17).

aMt contribute to a loss of the ecGCx, and increased EC mtROS could result in continued ecGCx shedding or impairment of its regeneration due to the activation of matrix metalloproteinases [162,163,164]. The co-occurrence of aMt, increased mtROS, and ecGCx shedding could be bidirectional, such that a vicious cycle might ensue. Decreasing mtROS from leaky aMt results in improved homeostasis [112,121], and similarly restoring the ecGCx could improve homeostasis and function [148,149,150]. A bidirectional role between the accumulation of aMt (due to impaired mitophagy) and the dysfunction or loss of the ecGCx in ECs is definitely emerging (Figure 16 panel 2) [11,148,149,150].

Certainly, more research may be necessary in order to confirm this bidirectional relationship; however, it remains a very intriguing association and presents an emerging opportunity to further unlock some of the mysteries associated with aMt, obesity, IR, LR, hyperglycemia, and the MetS reloaded in addition to associated complications of diabetic end-organ disease. Notably, this bidirectional concept would allow for future interventions by utilizing SGLT2 inhibitors such as empagliflozin [130], the Mt carbonic anhydrase inhibitor topiramate [122], uncoupling proteins such as UCP2 [165], and the emerging role of Mt transfer [166,167,168].

In order to supplement the restoration of the dysfunctional, attenuated, or loss of the ecGCx as occurs in sepsis, COVID-19, neuroinflammation and neurodegenerative disease, and T2DM, one would need to make provisions for an adequate supply of sulfur compounds and/or thiols [151]. From this standpoint, sulodexide (SDX), a heparin sulfate-like compound resistant to degradation by heparanase, accelerated ecGCx regeneration in vitro and in vivo [169], sodium thiosulfate (STS) [136,170], and n-acetyl-cysteine (NAC) [171] are approved and available for clinical use and have been recently utilized in the treatment of the T2DM and sepsis, glia-mediated neuroinflammation in neurodegenerative disease, and COVID-19.

## 7. MetS Reloaded and Metainflammation

Notably, Hotamisligil et al. were the first group to demonstrate that the adipose-derived cytokine tumor necrosis factor alpha (TNFα) provided the first link between the adipose tissue and IR, and eventually linked adipose tissue metainflammation to the MetS [172,173].

Chronic low-grade sterile inflammation (metainflammation) may instigate the origin of the metabolic syndrome [174,175]. Predisposing factors such as physical inactivity, overnutrition, obesity, and aging could result in cytokine hypersecretion via visceral adiposity and adipocyte-derived adipokines associated with crown-like structures and increased adipose cytokine secreting macrophages. Furthermore, these factors could eventually lead to IR and T2DM in genetically or metabolically predisposed individuals, (as discussed in Section 1) according to Grundy et al., regarding metabolic susceptibility and obesity (Figure 1 and Figure 3) [2,31]. Additionally, gut microbiota dysbiosis is associated with excess nutrients, high fat, sucrose, and fructose Western DIO, MetS, IR, LR, prediabetes, and is manifest T2DM in preclinical diabetic *db*/*db and* BTBR *ob*/*ob* models. Gut dysbiosis is known to result in leaky gut-derived metainflammation as a result of gut microbiome dysbiosis and metabolic endotoxemia due to the leakage of LPS into the systemic circulation [174,175,176]. This dysbiosis is associated with a leaky gut due to the attenuated dysfunction and loss of the intestinal lining epithelial TJ/AJ that allows LPS and other PAMPs to enter the systemic circulation. These released PAMPS and LPS (derived from cellular epithelial membranes of the dysbiotic intestinal gram-negative bacteria) result in increased peripheral cytokines/chemokines that signal TLR-4 receptors. Once signaled, toll-like receptors (TLRs) signal *NF*-κB activation and downstream cytokines/chemokines to result in metabolic endotoxemia- metainflammation. This gut-derived endotoxemia and metainflammation are capable of macrophage activation within the VAT to become inflamed and present as microscopic crown-like structures (CLS) that secrete inflammatory cytokines/chemokines to cause VAT-derived obesity-related metainflammation [174,175,176]. Importantly, there may exist a triangulation of metabolic functions, gut microbiota, and the innate immune system (Figure 18) [177,178,179,180].

Notably, the author fondly remembers when Schmidt et al., presented their data regarding the importance of metainflammation in the development of T2DM via decreased albumin, increased fibrinogen, orosomucoids, and sialic acid in 1999 at the American Diabetes Association [181], and the follow-up study in 2003, which demonstrated elevations in interleukin-6 and C-reactive protein [182]. Thus, the combination of obesity and gut dysbiosis associated with obesity are important causative factors in the development of metabolic endotoxemia and metainflammation in the setting of the MetS reloaded and manifest T2DM [181,182,183].

In addition to the important role of IR, hyperleptinemia and selective leptin resistance (LR) play an important role in the development of metainflammation in the MetS [30,174,175,176,182,183]. Leptin is known to be proinflammatory, and in selective leptin resistance, hyperleptinemia could further enhance metainflammation. Selective leptin resistance (LR) implies that some of the signaling effects of leptin are impaired, while some, such as those providing for the heightened effect on inflammation, could be additive to the proinflammatory effects due to IR [184,185,186,187,188]. VAT-derived cytokine expression that includes the perivascular adipose tissue (PVAT) is another plausible mechanism for the metainflammation-MetS reloaded relationship (Figure 19 and Figure 20) [121,189].

Note the extruded contents of the ruptured adipocytes and macrophages in the preclinical *db*/*db* model (Figure 19C,F). These extruded vesicles appeared to be very uniform with variable electron-dense staining and vesicle-like morphology. These ruptured adipocytes and macrophage vesicles are thought to be small exosomes due to their nanosized diameter (60–70 nm), and exosomes are known to range from 50–100 nm in diameter, making these vesicles within the size range of definite small exosomes [121,189].

### 7.1. MetS Reloaded and Adipokines/Hormones, Cytokines/Chemokines, and Toxic Free Fatty Acids (FFA)

In obesity, the adipose tissue is an active endocrine and paracrine organ that releases several bioactive mediators including adipokines, cytokines/chemokines and toxic saturated free fatty acids (FFA) that influence not only body weight homeostasis but also inflammation, coagulation, fibrinolysis, as well as insulin and leptin resistance [190]. When viewing the vast amount of adipose tissue (both SAT and VAT) in the obese Zucker model in Figure 4, one can almost envision the adipokines, cytokines/chemokines, and toxic FFAs spewing forth from these expanded; excessive adipose depots. Indeed, adipokines, cytokines/chemokines, and FFAs each have various functions in obesity and effects on clinical CCVD (Figure 21) [190,191,192,193,194].

A more complete list of adipokines can be viewed in the following reference [195].

### 7.2. PVAT Adipocyte and Macrophage-Derived Exosome microRNAs (miRNAs) and Long-Non-Coding RNAs (lncRNAs)

PVAT adipocyte-derived extracellular vesicle small exosomes (PVAT-dEVexos) and PVAT macrophage-derived extracellular vesicle small exosomes (PVATMΦ-dEVexos) are capable of delivering a multitude of proteins, lipids, and nucleic acids (RNA and DNA) via micro RNAs (miRNAs) and long-non-coding RNAs (lncRNAs) (longer than 200 nucleotides). These RNAs are capable of signal transduction and even the exchange of genetic information [180]. Thus, the obese PVAT adipose-derived exosomes are an important source of circulating exosomes that are capable of regulating the gene expression of adjacent cells (paracrine function) or distant cells and tissues (endocrine function) via their miRNAs in an inter-organ signal transduction fashion. These exosome-like vesicles are known to be related to the development of IR via the toll-like receptor (TLR4)/Toll-interleukin-1 receptor (TIR) domain–containing adaptor protein inducing interferon-β (Trif)–(TLR4/TRIF) pathway [196]. These exosomes are capable of promoting macrophage polarization by inhibiting Kruppel-like factor 4. Subsequently, these M-1-like macrophages are capable of increasing systemic inflammation and contribute to metainflammation and metabolic abnormalities induced by obesity in the MetS reloaded [197]. Indeed, miRNAs found within adipocyte and MΦ-derived exosomes are not only novel but also an exciting and emerging field of research that will continue to grow and change over time [180,198,199].

Importantly, Kwan et al., have recently examined the role of miRNAs that are generated by the obese adipocytes within the VAT-PVAT depots [200]. They have now identified multiple miRNAs that are capable of being shed from these obese adipocytes that have a paracrine cell-cell signaling mechanism to both interstitial macrophages and mast cells within these depots, in addition to providing an endocrine and inter-organ signaling mechanism to affect other organs such as the liver, cerebrocardiovascular tissues, and pancreatic islet β-cells to promote insulin resistance and end-organ dysfunction. These miRNAs include the following: decreased miR-148 and miR-4269; increased miR-23b and miR-4429, which activates the TGF-β and Wnt/ β-catenin signaling pathways that promote fibrosis; increased miR-155 and miR-223 and decreased miR-34a that promote macrophage polarization to an M1-like phenotype; increased miR-34a promotes hepatic steatosis, glucose intolerance and IR; decreased miR141-3p; increased miR-222 and miR-27a promotes peripheral IR; and increased miR-221-3p promotes vascular remodeling [200]. Furthermore, Żbikowski et al., have identified key miRNAs generated from obese VAT-PVAT adipocytes and polarized M1-like macrophages that are capable of paracrine, endocrine, and inter-organ signaling that contributes to peripheral IR. They found the following increased miRNAs that are important in promoting IR: miR-27a, miR-29a, miR-141-3p, miR-155, and miR-222. Additionally, increased miR-34a and miR-155 promote the polarization of MΦs to M1-like macrophages [201]. Various miRNA profiles are beginning to emerge that implicate the PVAT adipocytes and macrophages to result in IR. Moreover, in due time, there will be certain miRNA profiles that will emerge to better define the MetS reloaded. Interestingly, the exosomal miRNA numbers may be somewhat analogous to the letters of the alphabet, in that they will eventually allow us to spell words with these numbers and eventually write sentences, paragraphs, and stories in regard to the various miRNAs profiles that are still emerging. In addition, investigations of the mechanisms of action of adipocyte and MΦ exosomal miRNAs and their role in the MetS, CCVD and T2DM will be of considerable value for a more thorough understanding of the pathobiological mechanisms of the MetS and its increased risk for CCVD and T2DM. Indeed, there are cell-specific miRNAs secreted by the VAT-PVAT adipocytes and activated, CLS, M1-like MΦs in obesity (Figure 22) [200,201,202,203].

lncRNAs are even more in their infancy with regard to the MetS; however, they are emerging rapidly and will soon contribute to our understanding, since they, like miRNAs, are transported via adipocyte and MΦ extracellular vesicle exosomes that may impact neuroglia activation and function including impaired cognition and neurodegeneration [204,205], as well as other organs throughout the body. These lncRNAs have the capability to control pre-mRNA processing, the transport of mature mRNAs to specific cellular compartments, the regulation of mRNA stability, protein translation; turnover, and gene transcription [205]. Additionally, lncRNAs are known to be involved in nuclear chromatin remodeling, which implies the possibility of post-natal epigenetic mechanisms. It is worthy of note that we have observed chromatin remodeling consisting of chromatin condensation in our TEM studies of microglia and oligodendrocytes in the obese, IR, hyperleptinemic, and diabetic *db*/*db* models [115,116].

Thus, there are three possible mechanisms that may be in play when evaluating the obese PVAT—VAT depots containing hypertrophic adipocytes, prone to rupture unilocular WAT adipocytes, and polarized M1-macrophages of the CLS. The first possible mechanism is the engorged unilocular adipocytes of the WAT PVAT-VAT that rupture and shed their toxic contents of FFA, oxidized lipids, ceramides, and exosomes to evoke an innate immune response. This innate immune response will result in an accumulation of macrophages forming CLS and incite both mast cell and MΦ activation to result in metainflammation with the production of both local and systemic damaging cytokines and chemokines. The second possible mechanism is the obese adipocytes and activated proinflammatory polarized M1-like macrophages which secrete miRNAs that promote peripheral insulin resistance, which results in remodeling changes to end-organs affected by both obesity and T2DM. The third possible mechanism is the elevated proinflammatory adipokine/hormone leptin, and the associated decrease in the protective anti-inflammatory adipokine/hormone adiponectin will result in the activation of innate PVAT immune cells such that they could develop excessive damaging cytokine/chemokine release that would be taken up by the peripheral circulation and promote systemic metainflammation, which we know interferes with insulin sensitivity and promotes IR and thus ties LR to IR in the MetS reloaded. These three mechanisms may act individually; however, they are most likely functioning in concert and synergistically. Importantly, these three scenarios complement the findings proposed by Chadt et al., in that they proposed three hypotheses to help bridge the gap between epidemiology and pathobiological mechanisms in obesity. Their three hypotheses included (1) The inflammation hypothesis or metainflammation hypothesis; (2) The adipose tissue expandability or overflow hypothesis, which predicts that obesity may result in increased ectopic lipid deposition and IR in insulin-sensitive tissues such as the skeletal muscle, cardiac muscle, liver, islet pancreatic *β*-cells, and the PVAT (VAT) itself; (3) The adipokine hypothesis, which refers to the WAT cells in the SAT to function as an endocrine organ with excessively increased leptin and decreased adiponectin via both autocrine, paracrine, and endocrine mechanisms to allow for the metabolic impairment of insulin-sensitive target tissues to develop IR, which associates with LR and eventually results in *β*-cell dysfunction and failure due to glucolipotoxicity with ensuing T2DM [206].

## 8. Conclusions

The MetS reloaded supports the concept that many novel risk factors and emerging variables now belong to the original MetS, as proposed by NCEP (2002) and Grundy et al. (2004) (Figure 1) [1,2] and multiple associations and organizations. This overview has attempted to validate novel emerging risk factors and variables as well as exploring each of the original and traditional four arms of the MetS reloaded, including hyperlipidemia, hyperinsulinemia and hyperamylinemia, HTN, and hyperglycemia in Section 2, Section 3, Section 4 and Section 5. Section 6 explored the role of EC activation—dysfunction and the ecGCx, and Section 7 examined the role of metainflammation in the MetS reloaded and the potential role of adipokines, cytokines/chemokines, toxic FFF, and PVAT-derived adipocyte and activated MΦ exosomes (Figure 1). It is hoped that the examination of the MetS reloaded will provide not only a better understanding of the original definition of metabolic syndrome, but will also allow us to keep the concept of the MetS reloaded at the forefront of our research concepts and clinical practices. After all, new concepts may fuel the fire for new findings.

There has been controversy over the years since early definition of the MetS and whether or not its whole may be greater than the sum of its parts [207,208,209,210]. The author has the following conclusions from this overview: Even if an individual does not meet the criteria for the diagnosis of the MetS (due to missing just a few under-evaluated measurements or mg/dl deficit cut points), and if there are other risks factors within the MetS criteria that are elevated, that these individual elevated risk factors should be treated aggressively if at all possible. Furthermore, clinicians should definitely evaluate and treat all T2DM and CCVD risk factors without regard to whether the individual meets the three out of the five criteria for the diagnosis of the metabolic syndrome [210]. For a balanced view of the MetS controversy written in 2007 that still holds true today, it is suggested that one read a very elegantly written manuscript by Ferrannini [211].

The MetS may certainly be viewed as a risk syndrome that definitely provides a platform for individual identification and multifactorial risk reduction in regards to its treatment profile, including the all-important necessary lifestyle changes, especially in regard to diet and exercise. Sometimes a simple acronym such as (A-FLIGHT-UR) is helpful in remembering the multiple risk factors, variables, and emerging novel metabolic toxicities and signaling mechanisms that can be applied to develop the individual risk reduction of each of the multiple abnormal risks and variables associated with the MetS reloaded (Figure 23) [12].

Each of these metabolic risk factors and variables promotes ROS—RONSS either directly or indirectly, and results in excessive oxidative and redox stress. Importantly, remember that ROS begets ROS, and ROS begets metainflammation, and metainflammation begets ROS and is capable of creating vicious cycles that may be referred to now as ROS-induced ROS release (RIRR) [129].

Multiple efforts have been made to utilize supportive ultrastructural TEM images and illustrations to discuss ultrastructural cellular remodeling and functional changes in the visceral adipose tissue depots including the PVAT, pancreatic islets, brain, and depict aMt, and activated ECs in the obese, MetS, IR, LR, prediabetic, and diabetic preclinical rodent models. This overview has also shared the important role of identifying the MetS reloaded with its multiple traditional risk factors and the evolving novel risk factors and variables as early as possible. The MetS has been embedded in controversy with regard to its importance and necessity since its initial definition. However, the term MetS and its definitions have persisted on a global scale for the past three decades, and it is likely to continue to persist as the MetS into the future.

In summary, the clinical utility of the MetS (International Classification of Diseases, Tenth Revision Clinical Modification diagnostic code (ICD-10-CM E88.81) is important. The early diagnosis of the MetS (E88.81) allows both healthcare providers and affected individuals to develop treatment plans for each of the specific risk factors and variable components and lifestyle changes. Furthermore, the early identification and treatment of the MetS allow the opportunity to prevent the development of CCVD and T2DM and their long-term complications concerning increased morbidity and mortality.

## Figures and Tables

**Figure 1 medicina-59-00561-f001:**
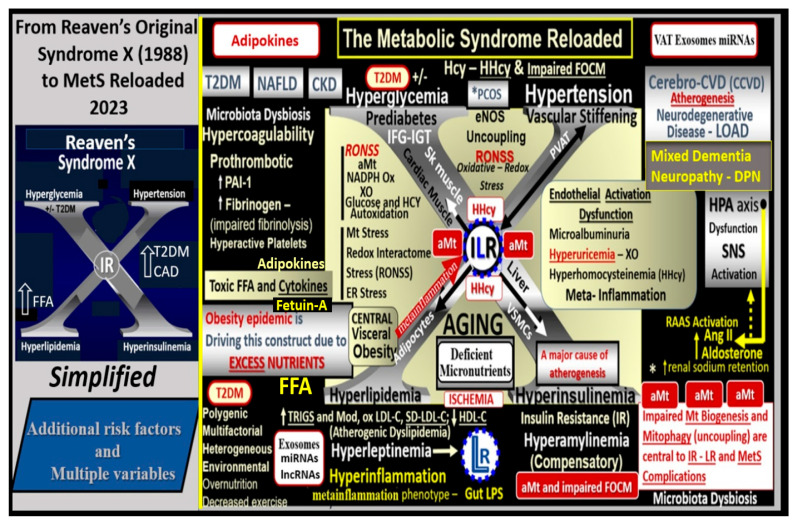
From Reaven’s simplified syndrome X to the complicated metabolic syndrome (MetS) reloaded. This image illustrates the simple Reaven’s Syndrome X (far left) to the complex MetS reloaded and its multiple risk factors. According to Reaven, the letter “X” was originally chosen because at the time coronary artery disease (CAD) and/or cerebrocardiovascular disease (CCVD) was a relatively unknown risk of human insulin resistance. Syndrome X was later termed the MetS by the National Cholesterol Education Program Adult Treatment Panel III and Grundy et al., 2002 and 2004, respectively. Hyperlipidemia (lower left arm), hyperinsulinemia, and hyperamylinemia (lower right arm), hypertension (essential) (upper right arm), and hyperglycemia with or without manifest T2DM (upper left arm) in the MetS reloaded represent the four arms to the central capital letter X. Importantly, visceral or central obesity is thought to be a major driver of this syndrome and is related to the emergent science of visceral adipose tissue (VAT) with adipocyte-derived adipokines and macrophage-derived cytokines/chemokines. Additionally, it is important to note the novel adipocyte and macrophage-derived exosomes with their novel signaling microRNAs (miRNAs) and long-non-coding RNAs (lncRNAs), which are capable of both paracrine and endocrine (inter-organ and long-distant) signaling in addition to the signaling via peripheral cytokine/chemokine adipokine network due to excessive meta-inflammation within VAT depots. Note that insulin resistance (IR) and leptin resistance (LR or Lr) (ILR) are placed centrally within the letter X. Importantly, note the red bold arrow connecting central visceral obesity to the central ILR within the letter X that is related to increased meta-inflammation. Also, note how the aberrant mitochondria (aMt) (a constant finding in obese and diabetic models) and hyperhomocyteinemia (HHcy) are flanking the central insulin and leptin resistance (ILR) that reflects impaired folate-mediated one-carbon metabolism (FOCM). Additionally, note the important role of microbiota dysbiosis and its emerging role associated with obesity and the MetS reloaded. Note the relationship between the HPA axis and the RAAS with increased sodium retention (*).Ang *II = angiotensin II*; *CAD = coronary artery disease*; *CKD = chronic kidney disease*; *CCVD = cerebrocardiovascular disease*; *DPN = diabetic peripheral neuropathy*; *eNOS = endothelial nitric oxide synthase*; *ER = endoplasmic reticulum*; *FFA = free fatty acids*; *Hcy = homocysteine*; *ILR = insulin/leptin resistance*; *LOAD = late-onset Alzheimer’s disease*; *lncRNAs = long non-cording RNAs*; *HPA = hypothalamic–pituitary–adrenal axis*; *LPS = lipopolysaccharide*; *LR = leptin resistance—selective Lr*; *miRNAs = micro ribonucleic acids*; *Mt = mitochondria*; *NADPH Ox = nicotinamide adenine dinucleotide phosphate oxidase*; *NAFLD = non-alcoholic fatty liver disease*; *NO = nitric oxide*; *Non-HDL-C = non-high-density lipoprotein-cholesterol*; *O2 = oxygen*; *oxLDL-C = oxidized low density lipoprotein-cholesterol*; *PAI-1 = plasminogen activator inhibitor -1*; *PCOS = polycystic ovary syndrome*; *RAAS = renin-angiotensin-aldosterone-system*; *RNA = ribonucleic acid*; *RONSS = reactive oxygen, nitrogen, sulfur species*; *RSI = reactive species interactome*; *sdLDL-C = small dense low-density lipoprotein-cholesterol*; *SNS = sympathetic nervous system*; *Trigs = triglycerides*; *XO = xanthine oxidase*.

**Figure 2 medicina-59-00561-f002:**
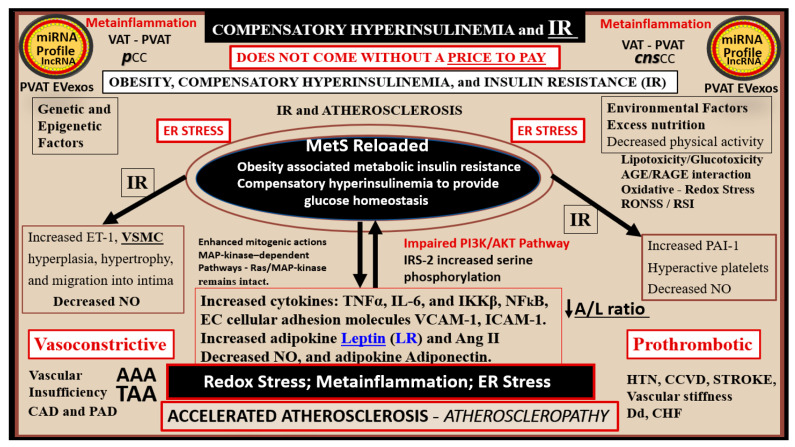
Insulin resistance (IR) and compensatory hyperinsulinemia do not come without a price to pay. IR plays a central role in the MetS and results in an impaired PI3K/AKT pathway due to impaired IRS-2 function (IRS-2 increased serine phosphorylation) and decreased insulin-stimulated NO synthesis and impaired vasodilation. However, the elevated insulin can still signal through the mitogenic MAPkinase-dependent pathway and contribute to arterial vessel wall remodeling and accelerated atherosclerosis (atheroscleropathy) due to pro-atherosclerotic, pro-thrombotic, and vasoconstrictive effects). The increased redox stress and inflammatory vicious cycle that drive IR are promoted by an associated increased peripheral and central nervous cytokines and chemokines (***p***CC and ***cns***CC respectively) and the IR promoting effects from adipose-derived miRNAs via increased extracellular vesicle exosomes (EVexos) and their microRNA profiles along with the ongoing metainflammation. *AAA = abdominal aortic aneurysm*; *Ang II = angiotensin II*; *CAD = coronary artery disease*; *CHF = congestive heart failure*; *ET-1 = endothelin-1*; *ER = endoplasmic reticulum*; *HTN = hypertension*; *ICAM-1= intercellular adhesion molecule 1*; *IKKβ = inhibitor of nuclear factor kappa-B kinase subunit beta*; *IL-6 = interleukin-6*; *lncRNA = long non-coding ribonucleic acid*; *LR = leptin resistance*; *miRNA = micro ribonucleic acid*; *NFkB = nuclear factor kappa-light-chain-enhancer of activated B cells*; *NO = nitric oxide*; *PAD = peripheral arterial disease*; *PAI-1 = plasminogen activator inhibitor-1*; *ROS = reactive oxygen species*; *RNA = ribonucleic acid*; *RONSS = reactive oxygen, nitrogen, sulfur species*; *RSI = reactive species interactome*; *TAA = thoracic aortic aneurysm*; *TNFα = tumor necrosis factor alpha*; *VCAM-1 = vascular cell adhesion molecule-1*; *VSMC = vascular smooth muscle cell(s)*.

**Figure 3 medicina-59-00561-f003:**
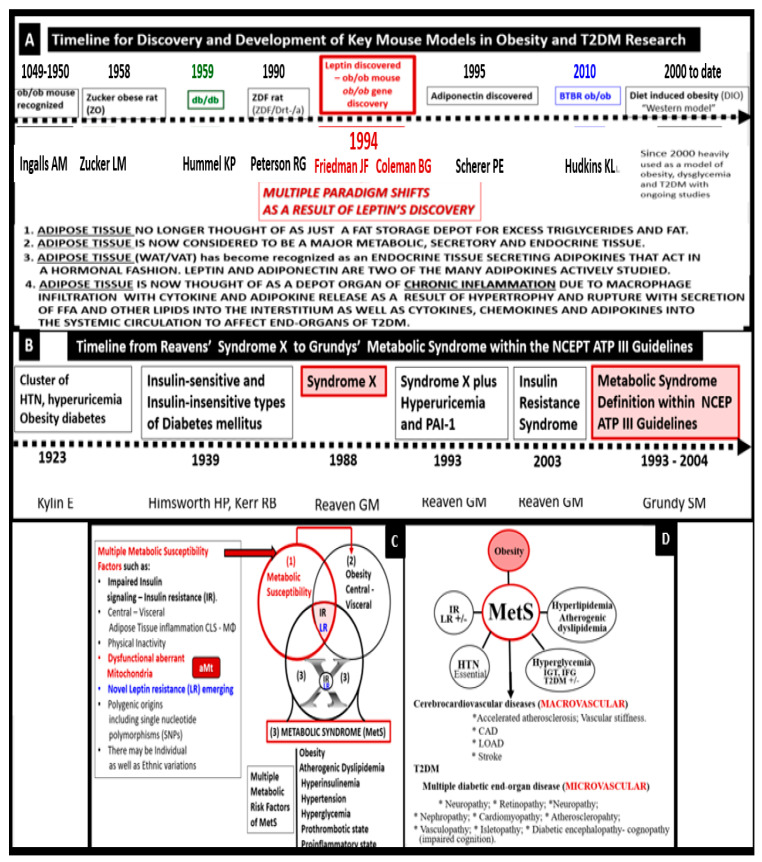
**Panel A** demonstrates the timeline for the development of key obesity and T2DM models from 1948 to date that have resulted in multiple paradigm shifts in regard to the metabolic syndrome (MetS). **Panel B** demonstrates the timeline of Reaven’s syndrome X to the NCEP ATP III guidelines for the MetS and Grundy’s emphasis on the Met S. These two combined timelines allow one to compare the importance of obesity, insulin and leptin resistance to the timeline for the development of the MetS. **Panel C** illustrates that the MetS is associated with multiple variables and the importance of an individual’s metabolic susceptibility factors (boxed-in factors) to better understand why only some obese individuals develop MetS. **Panel D** depicts that the MetS is associated with multiple variables and an increased risk for cerebrocardiovascular disease (CCVD) via macrovascular disease, while diabetic end-organ disease is associated with type 2 diabetes mellitus (T2DM) and microvascular disease. Panel A is adapted and modified with permission by CC 4.0 [30]. *FFA = free fatty acids*; *VAT = visceral adipose tissue*; *WAT = white adipose tissue*.

**Figure 4 medicina-59-00561-f004:**
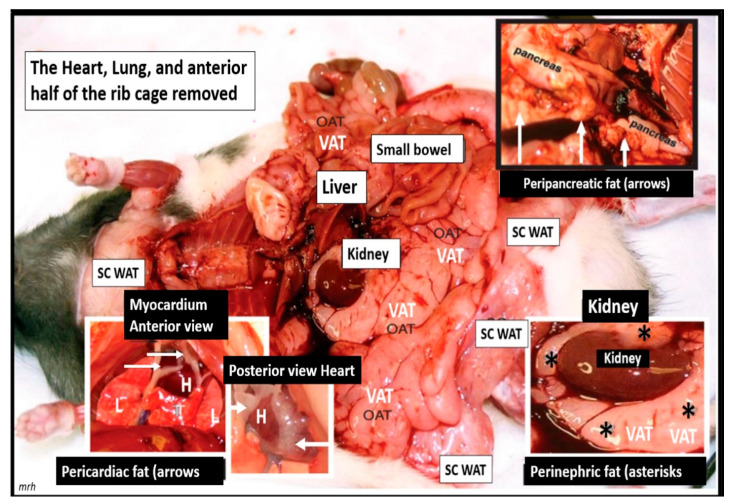
Necropsy of a male Zucker *fa*/*fa* or *ob*/*ob* at nine-weeks of age. Gross findings at necropsy of obesity in the nine-week-old adolescent male Zucker obese *ob*/*ob* rat model. This image depicts the massive accumulation of omental adipose tissue (OAT) or visceral adipose tissue (VAT) in the Zucker *fa*/*fa* or *ob*/*ob* model. Also, note there is an accumulation of excessive subcutaneous (SC) adipose tissue—white adipose tissue (WAT) depot. The accumulation of peripancreatic VAT fat (white arrows) (upper-right), pericardiac VAT fat (white arrows) (lower-left), and extremely excessive perinephric VAT fat (asterisks) (lower-right) are depicted in the inserts. This image is adapted with permission from open access [40]. *H = heart*; *L = collapsed lung*; *mrh = necroscopy-photographer initials melvin ray hayden*; *OAT = omental adipose tissue*; *T= thymus*.

**Figure 5 medicina-59-00561-f005:**
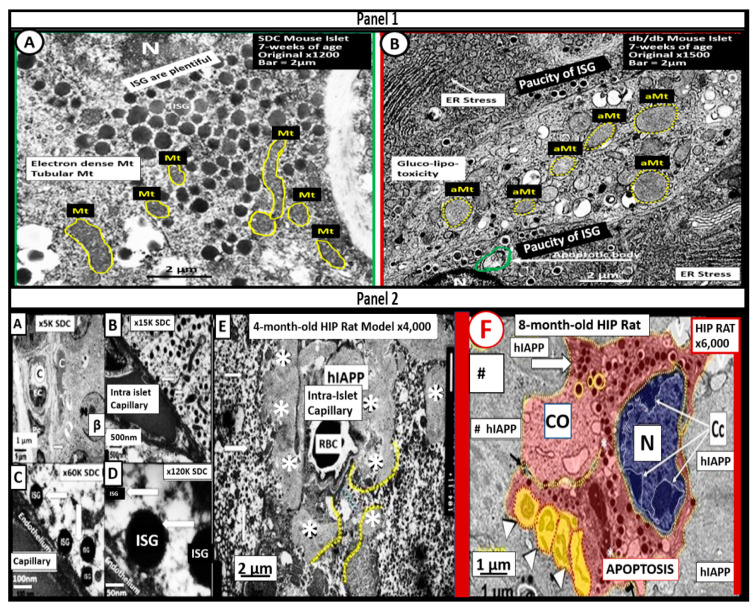
Ultrastructural remodeling of pancreatic islets and islet Beta-cells (β-cells) in obese and type 2 diabetes mellitus (T2DM) rodent models. **Panel 1** depicts the paucity of insulin secretory granules (ISGs), endoplasmic reticulum (ER) stress (arrow), and an increase in aberrant mitochondria (aMt) in the obese, diabetic *db*/*db* mouse (**B**) as compared to controls (**A**) **Panel 2** depicts the loss of ISGs and the excessive deposition of islet amyloid in the human islet amyloid polypeptide (HIP) (* in Panel 2E and # in Panel 2F) rat model (**E**) that results in β-cell apoptosis (**F**) as compared to controls (**A**–**D**) Note the nucleus chromatin condensation (arrows) in the apoptotic β-cell in panel F. Scale bars vary and are present. Images are reproduced and modified with permission by CC 4.0 [52].

**Figure 6 medicina-59-00561-f006:**
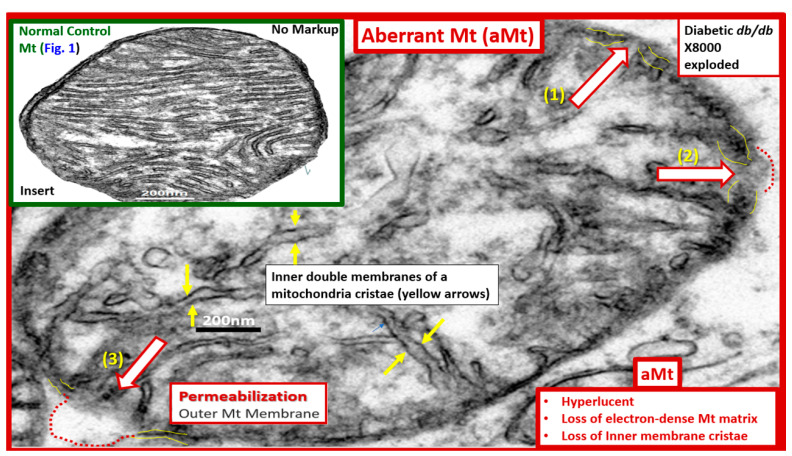
Aberrant mitochondria (aMt) in the MetS Reloaded. This image depicts an aberrant mitochondrion (aMt) from an, insulin-resistant, diabetic *db*/*db* female mouse model at 20 weeks of age. Aberrant mitochondria are a common phenotype found in multiple different cells in models of obesity, insulin resistance, metabolic syndrome, and type 2 diabetes mellitus as compared to control models such as the C47BL6J model demonstrated in the upper left-hand side of this image (Figure 1). Thinning with interruption and permeabilization of the outer aMt membrane is a common finding in obese, insulin-resistant, diabetic *db*/*db* models. This aMt depicts (**1**) the interruption of the inner and outer membranes with partial permeabilization (open red arrow and red dashed line); (**2**) the thinning of the inner and outer membranes (open red arrow); and (**3**) the complete loss of the inner and outer membranes with permeabilization of the outer membrane (open red arrow, dashed red lines. Increased permeabilization of the aMt outer membrane allows for the leakage of multiple mitochondrial-derived toxicities including mitochondrial-derived reactive oxygen species (Mt-ROS), cytochrome c, and proinflammatory MtDNA. This image is reproduced and adapted with permission by CC 4.0 [11]. Original magnification x2000; scale bar = 200 nm.

**Figure 7 medicina-59-00561-f007:**
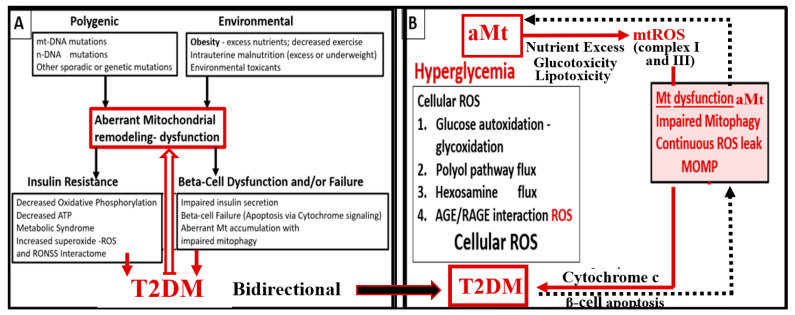
Type 2 diabetes mellitus (T2DM) is a multifactorial and polygenic disease in addition to having dysfunctional aberrant mitochondria (aMt) capable of bidirectional signaling. **Panel A** depicts that aberrant mitochondrial (aMt) phenotypes are a central and critical defect that leads to T2DM via multifactorial polygenic and environmental factors. **Panel B** demonstrates that aMt dysfunction with increased mitochondrial reactive oxygen species (mtROS) and T2DM are capable of interacting via bidirectional mechanisms. *ATP = adenosine triphosphate*; *AGE= advanced glycation end product*; *mt-DNA = mitochondrial deoxyribonucleic acid*; *MOMP = mitochondria outer membrane permeabilization*; *n-DNA = nuclear DNA*; *RAGE = receptor for AGEs*; *ROS = reactive oxygen species*; *RONSS = reactive oxygen, nitrogen, sulfur species*.

**Figure 8 medicina-59-00561-f008:**
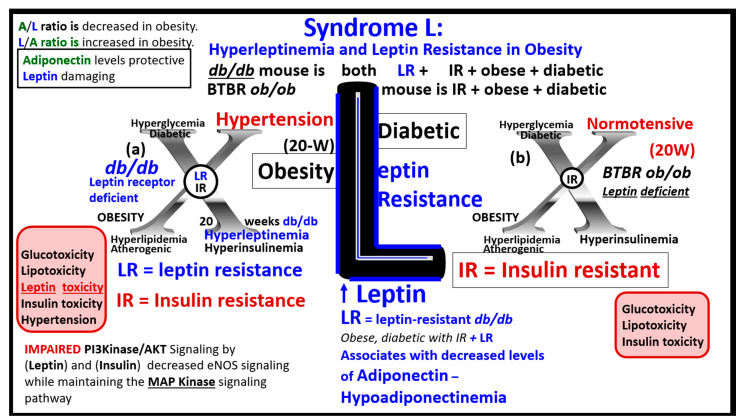
Syndrome L—hyperleptinemia and selective leptin resistance (LR) are emerging and novel risk factors in obesity, insulin resistance (IR), metabolic syndrome (MetS), and type 2 diabetes mellitus (T2DM). This illustration compares the leptin receptor deficiency *db*/*db* with both the leptin resistance (LR) panel (**a**) to the leptin deficiency BTBR *ob*/*ob* mouse models with IR that develops obesity and the diabetes panel (**b**) at 20 weeks (W) of age. LR is known to lead to glucose intolerance at least in the Zucker obese *fa*/*fa* rat models, *db*/*db* mouse models, and occurs primarily due to hepatic glucose overproduction that may occur even prior to developing obesity. LR is an emerging and novel finding in the MetS reloaded, as previously depicted in Figure 1. Importantly, its relationship to the MetS reloaded has increased over recent years. Leptin resistance is associated with decreased adiponectin and is proinflammatory, proatherosclerotic, and associates with decreased insulin sensitivity. *AKT = protein kinase B*; *MAP Kinase = mitogen-activated protein kinase*; *PI3 Kinase = phosphoinositide 3-kinase*.

**Figure 9 medicina-59-00561-f009:**
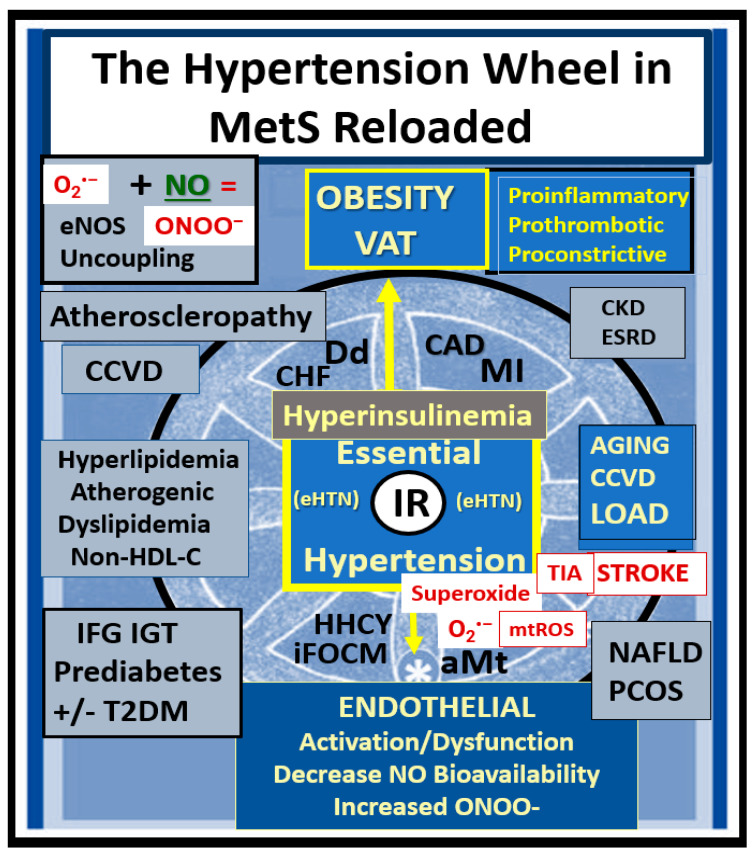
The essential hypertension (HTN) wheel in the metabolic syndrome (MetS) reloaded. This wheel depicts that insulin resistance (IR) is a central core feature of the wheel as it is in the MetS reloaded. IR is a central mediator for the development of HTN and IFG-IGT–T2DM, along with multiple metabolic and clinical boxed-in conditions that surround the outer portions of this hypertension wheel. Obesity is placed at the top of the wheel because it is believed to be the driving force behind the subsequent clinical end-organ remodeling and disease and the development of HTN. Endothelial activation and dysfunction (bottom of the wheel) result in increased peroxynitrite (ONOO^−^) and decreased nitric oxide (NO) bioavailability. Peroxynitrite is generated by the reaction between superoxide and NO. One can note the multiple diseases that are associated with HTN, and the wheel depicts the interconnectedness between HTN and the multiple disease states, including vascular stiffening. Thus, IR, HTN, and T2DM are not to be underestimated. Atheroscleropathy is a term that may be used when discussing accelerated atherosclerosis and macrovascular disease in those individuals with T2DM and the MetS reloaded. The wheel was chosen as a background icon because it goes round and round, and over time it just keeps on turning and results in vascular stiffening and end-organ damage in the heart-brain-kidney axis that has high capillary flow with low resistance and increased vulnerability to the increased pulse wave velocity associated with vascular stiffening, HTN, and microvascular disease. *CAD = coronary artery disease; CCVD = cerebro-cardiovascular disease; CHF = congestive heart failure; CKD = chronic kidney disease; Dd = diastolic dysfunction; eNOS = endothelial nitric oxide synthase; ESRD = end-stage renal disease; IFG = impaired fasting glucose; IGT = impaired glucose tolerance; LOAD = late-onset Alzheimer’s disease; MI = myocardial infarction; mtROS = mitochondrial reactive oxygen species;* O_2_^•–^
*= superoxide*.

**Figure 10 medicina-59-00561-f010:**
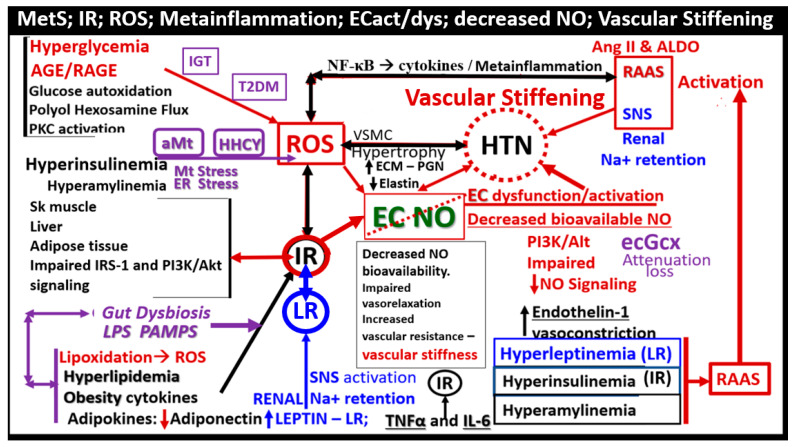
Pathophysiology of essential hypertension (HTN) in the MetS reloaded. This figure demonstrates the complex interconnected pathways between insulin resistance (IR encircled in red color) and HTN. IR is the cornerstone of the MetS reloaded, and this schematic depicts how it is related to the development of HTN (dashed red circle). Note the important role of endothelial derived nitric oxide (NO) (green), which is essential for proper vascular relaxation and homeostasis, and how it is negated (red-dashed diagonal line) in the development of HTN. Importantly, decreased bioavailable NO results in decreased vasorelaxation and increased vascular resistance with a resulting increase in vascular stiffening and elevated blood pressure (BP). Next, note the devastating role that reactive oxygen species (ROS) has on NO and endothelial cell dysfunction and activation, resulting in decreased vasorelaxation, increased vascular (Vasc) resistance, blood pressure elevation, vascular stiffness, and the role they each play in the development of HTN. Activation of the renin-angiotensin-aldosterone system (RAAS) is also very important to the development of HTN as well as the activation of the sympathetic nervous system (SNS). Importantly, note the involvement of the Leptin—Leptin resistance (LR) pathway in blue coloring. Furthermore, note the emerging roles of the endothelial glycocalyx (ecGCx), aberrant mitochondria (aMt), hyperhomocysteinemia (HHCY), and gut dysbiosis, which are related to inflammation via lipopolysaccharide (LPS), as these are emerging roles not only of the MetS reloaded but also in the development of HTN (colored in purple). The sum of these interacting pathways and functional changes are associated with vascular wall remodeling and vascular stiffening. This image adds additional information regarding some of the possible mechanisms in the development of HTN presented in figure nine because it focuses more on the involved possible mechanistic pathways in the development of HTN. Not shown is the relationship between the sodium/potassium ATPase enzyme that is redox-sensitive and the excessive redox stress in the MetS that is currently thought to also promote a salt-sensitive HTN due to the sodium/potassium ATPase enzyme inhibition. While some of these mechanisms may seem redundant, they are actually complementary to Figure 9. The inverse relationship between leptin and adiponectin also plays an important role, because when leptin is elevated, adiponectin is decreased, and thus there is a decrease in the A/L ratio that is associated with the loss of adiponectin’s protective vascular role. Importantly, systolic HTN predominates in the MetS reloaded. *AGE/RAGE = advanced glycation end products and receptor for AGE; Akt = protein kinase B; ALDO = aldosterone; Ang II = angiotensin II; ECM = extracellular matrix; IGT = impaired glucose tolerance; IL-6 = interleukin-6; LPS = lipopolysaccharide; Na + = sodium; NF-κB = nuclear factor kappa B; PAMP = pathogen associated molecular pattern; PGN = proteoglycan; PI3K = phosphoinositide 3-kinase; PKC = protein kinase C; Sk = skeletal; T2DM = type 2 diabetes mellitus; TNFα = tumor necrosis factor-alpha; VSMC = vascular smooth muscle cell*.

**Figure 11 medicina-59-00561-f011:**
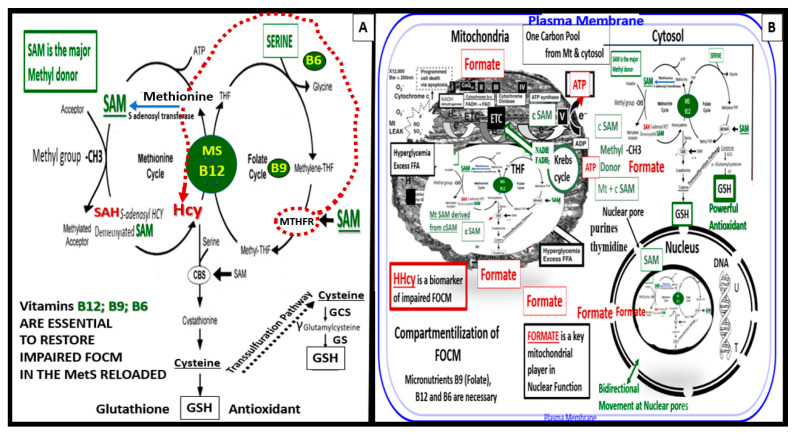
Folate-mediated one-carbon metabolism (FOCM). FOCM involves multiple complex interacting cycles within the cytosol, mitochondria, and nucleus in the metabolic syndrome (MetS) reloaded. **Panel A** demonstrates the methionine and folate cycles and supports the importance of the methyl donor S-adenosylmethionine (SAM), as well as demonstrating the importance of the essential B12, B9, and B6 vitamins. **Panel B** illustrates that FOCM is compartmentalized to the cytosol, mitochondria, and nucleus. Note that formate and SAM are transferred from the mitochondria to the nucleus via nuclear pores. This figure is modified and adapted with permission by CC 4.0 [17].

**Figure 12 medicina-59-00561-f012:**
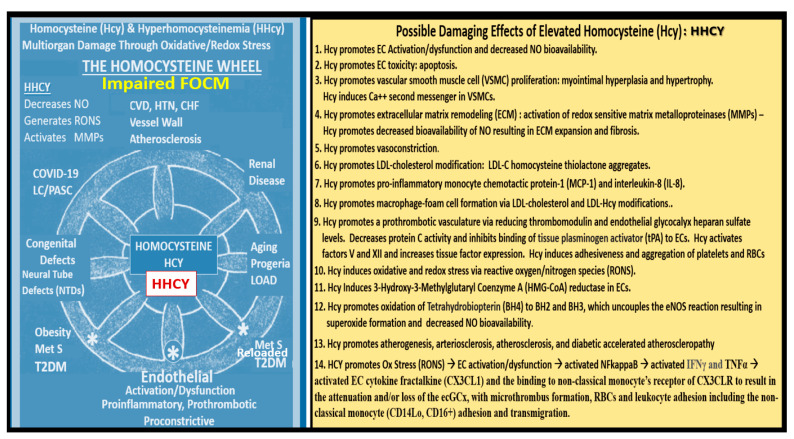
Impaired folate-mediated one-carbon metabolism (FOCM) and hyperhomocyteinemia (HHCY) are associated with multiorgan damage and multiple clinical diseases via excessive redox stress. Once impaired FOCM develops and goes unchecked, this homocysteine (HCY) wheel begins turning and just keeps on turning, causing damage to the various organs and tissue systems identified, and results in multiple clinical disease states depicted on the HCY wheel (left-hand side of this image). When impaired FOCM and HHCY develops, the elevated HCY is capable of undergoing autoxidation, the formation of Hcy mixed disulfides, the interaction of Hcy thiolactones, and protein homocysteinylation reactions that result in damage and dysfunction to proteins, lipids, and nucleic acids. Note that the asterisks indicate increased importance. These damaging effects result in at least 14 damaging effects enumerated on the right-hand side of this image. This image is provided with permission by CC 4.0 [17]. *CHF = congestive heart failure*; *COVID-19 = coronavirus disease-19*; *CVD = cerebro-cardiovascular disease*; *EC = endothelial cell*; *ecGCx = endothelial cell glycocalyx*; *eNOS = endothelial nitric oxide synthase*; *HTN = hypertension*; *IFNγ = interferon gamma*; *NO = nitric oxide*; *LDL = low-density lipoprotein cholesterol*; *MetS = metabolic syndrome*; *MMPs = matrix metalloproteinases*; *NFkappaB = nuclear factor kappa B*; *NTD = neural tube defects*; *LC/PASC = long COVID/* post-acute sequelae of SARS-CoV-2; *LOAD = late-onset Alzheimer’s disease*; *Ox = oxidative stress*; *RBCs = red blood cells*; *RONS = reactive oxygen nitrogen species*; *T2DM = type 2 diabetes mellitus*. *TNFα = tumor necrosis factor alpha*.

**Figure 13 medicina-59-00561-f013:**
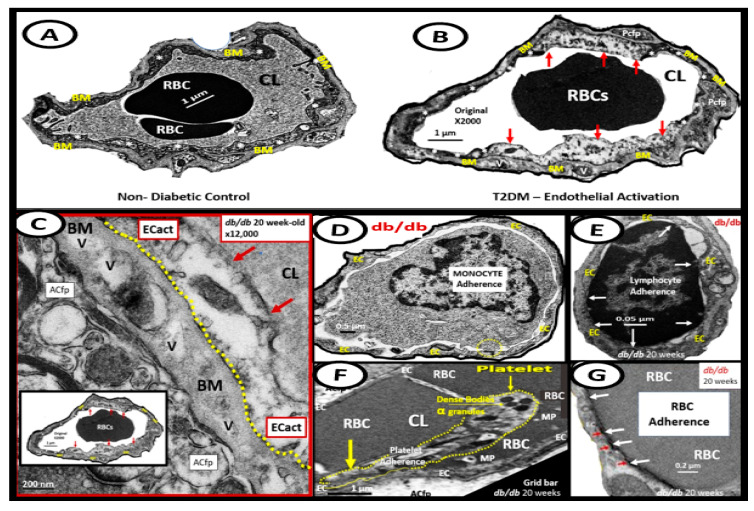
Examples of transmission electron microscopic (TEM) images for endothelial cell activation (EC*act*). **Panel B** depicts the thickened electron-lucent areas (red arrows) of EC*act* as compared to control models in **Panel A**. **Panel C** depicts basement membrane (BM) thickening with increased vacuoles (V) and vesicles (v). **Panels D and E** depict monocyte (**D**) and lymphocyte (**E**), platelet (**F**) and red blood cell (RBC) adhesion (**G**) to the activated ECs, respectively. Adhesions sites in panels **E** and **G** are denoted by white arrows. Original magnification x2000; scale bar = 1 μm, and images are modified with permission of CC 4.0 [55]. Images (**Panels C**–**G**) are reproduced and modified with permission of CC 4.0 [11,136]. Magnifications and scale bars vary. *ACfp = astrocyte foot processes*; *Cl, capillary lumen*; *EC = endothelial cells*; *ECact = endothelial cell activation*; *MP = microparticle of the platelet*.

**Figure 14 medicina-59-00561-f014:**
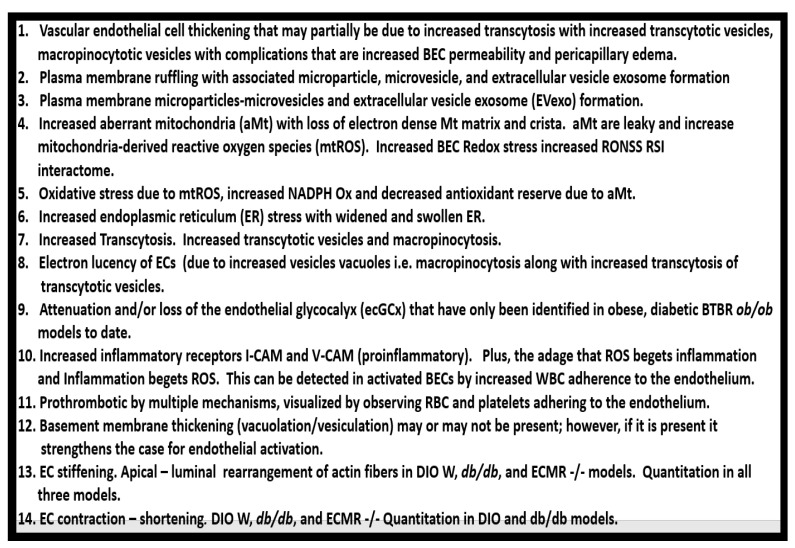
Summary of the observational ultrastructural transmission electron microscopic (TEM) remodeling changes in activated systemic and brain endothelial cells. *ECMR = endothelial cell mineralocorticoid receptor; DIO = diet induced obesity; EC(s) = endothelial cell(s); BEC(s) = brain endothelial cell(s); ER = endoplasmic reticulum; MtROS= mitochondria reactive oxygen species; NADPH Ox =nicotinamide adenine dinucleotide phosphate oxidase; RBC(s) = red blood cell(s); RONSS = reactive oxygen, nitrogen, sulfur species; ROS = reactive oxygen species; RSI = reactive species interactome; W = Western mice*.

**Figure 15 medicina-59-00561-f015:**
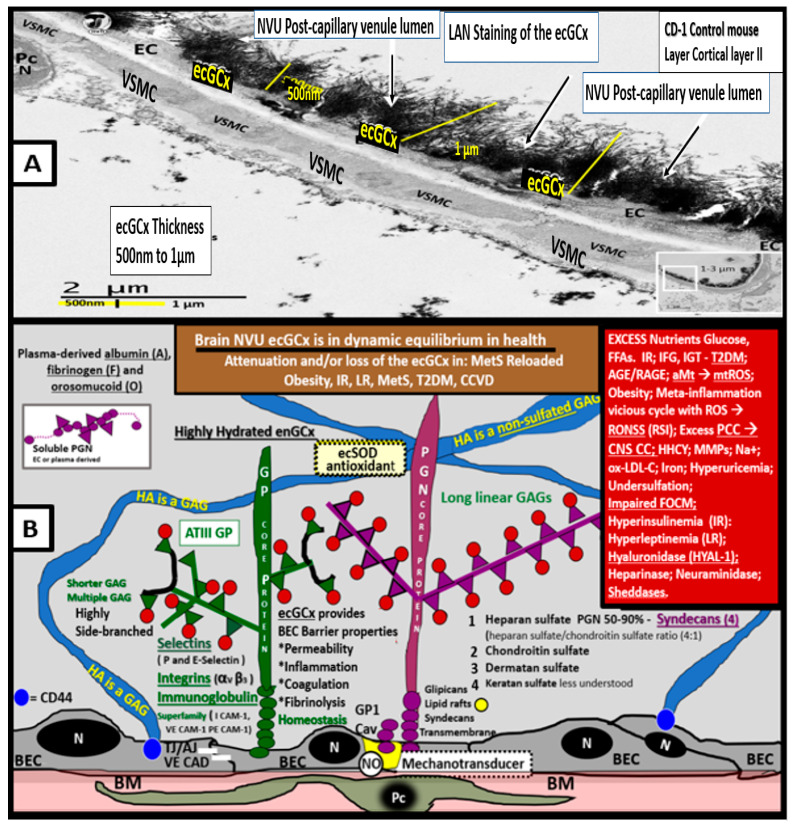
Lanthanum nitrate staining of the endothelial glycocalyx (ecGCx) and an illustration of the (ecGCx) in the metabolic syndrome reloaded. **Panel A** demonstrates the normal lanthanum nitrate (LAN) perfusion fixation staining of the endothelial glycocalyx (ecGCx) in a post-capillary venule. This exploded image demonstrates the normal staining of the ecGCx in a healthy control male CD-1 mouse brain from the frontal cortical grey matter layer III with the original scale bar of 2 μm intact. Note the intense electron-dense staining of lanthanum nitrate of the apical brain endothelial cell(s) (BECs) for the ecGCx such that one cannot visualize the structural content of the ecGCx that is revealed in the following panel B illustration. Note the boxed-in insert in the lower right-hand corner (scale bar = 2 μm) with a white outline, which is the original image from which the exploded image in panel A is derived. Scale bar (black) = 2 μm and yellow scale bars = 500 nm. **Panel B** illustrates the various proteoglycans (PGNs) purple, glycoproteins (GPs) green, hyaluronan (HA) blue, glycosaminoglycans (GAGs) purple and green triangles, and their sulfation sites (red circles). Note that the asterisks indicate increased importance. **Panel A** is an original image recently acquired by the author that has not been previously published, and is presented only for its educational purposes. Panel B is a modified and adapted image with permission of CC 4.0 [30]. *A = albumin; AGE/RAGE = advanced glycation end BM = basement membrane; CAD = cadherin; CAM = cellular adhesion molecule; CD44 = cluster of differentiation 44; EC = endothelial cell(s); ecSOD = extracellular superoxide dismutase; F = fibrinogen; FGF2 = fibroblast growth factor 2; FOCM = folate-mediated one-carbon metabolism; GCx = glycocalyx; ICAM*-1 *= intercellular adhesion molecule; Ox LDL = oxidized low-density lipoprotein; LPL = lipoprotein lipase; MMPs = matrix metalloproteinases; N = nucleus; Na+ = sodium; O = orosomucoids; Pc =vascular mural cell pericyte(s); PECAM- 1= platelet endothelial cell adhesion molecule-1. RONS = reactive oxygen species; VEC = vascular endothelial cell(s); TFPI = tissue factor pathway inhibitor; TJ/AJ = tight and adherens junctions; VCAM = vascular cell adhesion protein; VE CAD = vascular endothelial cadherins; VEGF = vascular endothelial growth factor; XOR = xanthine oxioreductase*.

**Figure 16 medicina-59-00561-f016:**
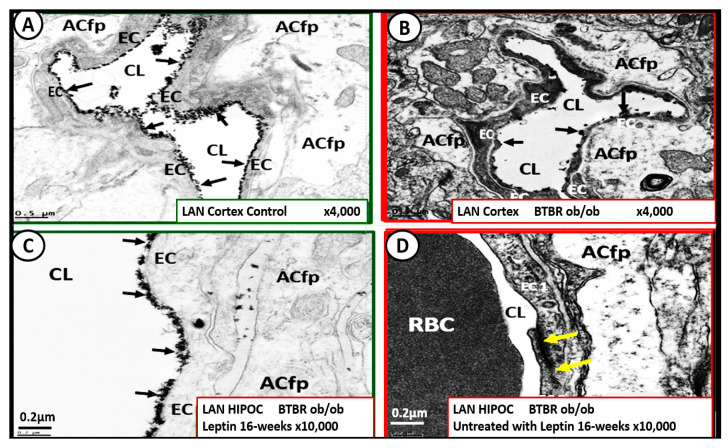
The obese diabetic BTBR *ob*/*ob* brain endothelial glycocalyx (ecGCx) in cortical layer III is protected with leptin replacement. This suggests that the presence of functioning leptin is important in maintaining a proper ecGCx covering of endothelial cells and that hyperleptinemia and selective leptin resistance (LR) could interfere with a healthy ecGCx as well as insulin resistance (IR). **Panel A** illustrates in the heterozygous non-diabetic control model cortical layer III (arrows) that the ecGCx is continuous. **Panel B** depicts a marked attenuation and/or loss of the ecGCx, and note that in regions where it is stained that it appears clumped (arrows) and discontinuous in the obese diabetic BTBR *ob*/*ob* model when compared to the control model in **Panel A**. **Panel C** displays a continuous coverage by the ecGCx in hippocampus CA-1 regions of the BTBR *ob*/*ob* models that were treated with intraperitoneal leptin for 16-weeks and stained with LAN (arrows). **Panel D** depicts the complete loss of LAN staining in the hippocampus CA-1 regions of the BTBR *ob*/*ob;* note that the tight and adherens junction (TJ/AJ) remains intact (yellow arrows). Importantly, the loss of the ecGCx may result in increased permeability. Magnification x4000; scale bar = 0.5 μm in **Panels A** and **B**. Magnification x10,000; scale bar = 0.2 µm in panels **C** and **D**. **The supplemental two-panel image** below the top four-panel (A–D) demonstrates longitudinal images of an intact ecGCx in control wild type (WT) models on the left-hand side, while there is a noticeable attenuation and loss of the ecGCx in the leptin-deficient BTBR *ob*/*ob* model on the right-hand side in hippocampal regions. Magnification x10,000; scale bar = 0.2 μm. Images provided and modified with the permission of CC 4.0 [30]. *ACfp = astrocyte foot process; Cl = capillary lumen; EC = brain endothelial cell; HC and HIPOC = hippocampus CA-1 regions*.

**Figure 17 medicina-59-00561-f017:**
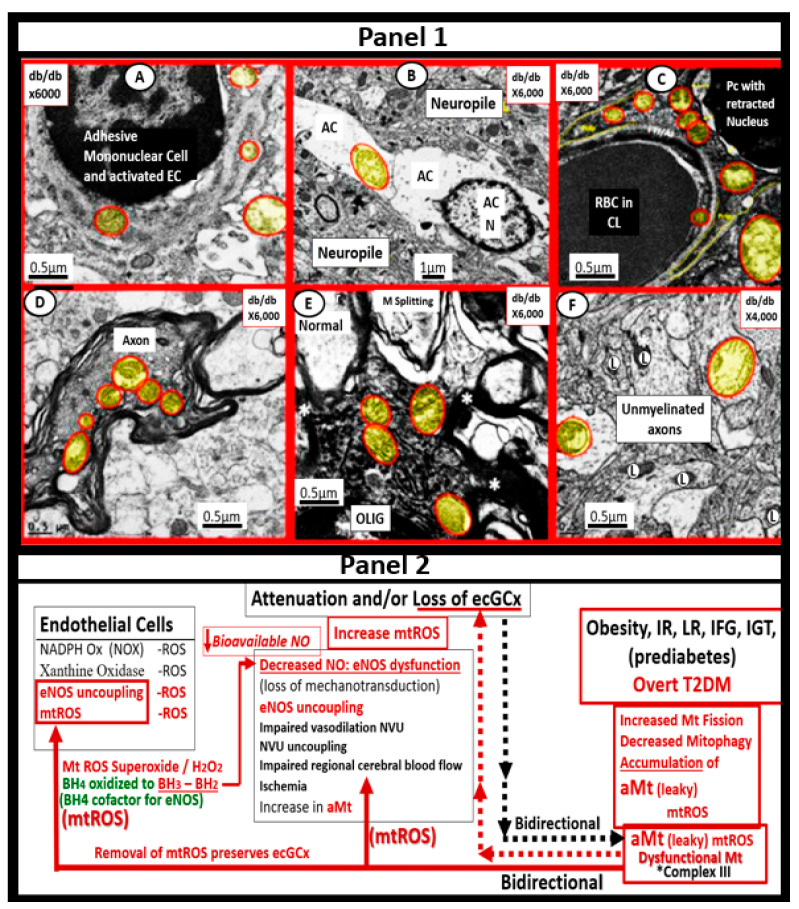
Multiple cell types in the brain contain aberrant mitochondria (aMt), and emerging evidence suggest a bidirectional relationship between aMt and the attenuation or loss of the endothelial glycocalyx (ecGCx). **Panel 1** depicts multiple cell types in the brain with aMt (pseudo-colored yellow with red outlines). These aMt are in contrast to the smaller electron-dense Mt found in control model brain endothelial cells. aMt are hyperlucent and lose their electron dense mitochondrial matrix and cristae in brain endothelial cells (BECs) (**A**), astrocytes (AC) (**B**), pericytes (Pc) and foot processes (f**C**), myelinated neuronal axons (**D**), oligodendrocyte (OLIG) (**E**), and unmyelinated axons (**F**). These modified images are provided by CC 4.0 [121,122,123]. Magnifications and scale bars are included in each panel. **Panel 2** illustrates that there may be a bidirectional relationship between aMt and dysfunction, attenuation, and/or loss (shedding) of the brain BECs ecGCx. This illustration establishes the role of oxidant stress–reactive oxygen species (ROS) in BECs and specifically mitochondrial ROS (mtROS) (left-hand box). This schematic also shows that obesity, IR, LR, impaired fasting glucose (IFG), impaired glucose tolerance (IGT), and overt T2DM are related to increased Mt fission, decreased mitophagy, and the accumulation of leaky aMt that leak mtROS (right-hand box). Leaky aMt may be responsible for the attenuation and/or loss of the ecGCx (red-dashed arrows), and that, in turn, may result in the loss of the ecGCx that may contribute to an increase in aMt (black-dashed arrows). In addition, mtROS superoxide or hydrogen peroxide (H2O2) could oxidize the essential tetrahydrobiopterin (BH4) cofactor that is absolutely essential for the eNOS enzyme to produce nitric oxide (NO) and result in eNOS uncoupling. eNOS uncoupling could ultimately result in decreased bioavailable NO. The depicted bidirectional interaction could result in a vicious cycle. This vicious cycle could be interrupted by either preventing the accumulation of aMt (improved mitophagy) or preventing the dysfunction, attenuation, and/or loss (shedding) of the ecGCx. *Asterisk = myelin (M); L = lysosome; RBC = red blood cell*.

**Figure 18 medicina-59-00561-f018:**
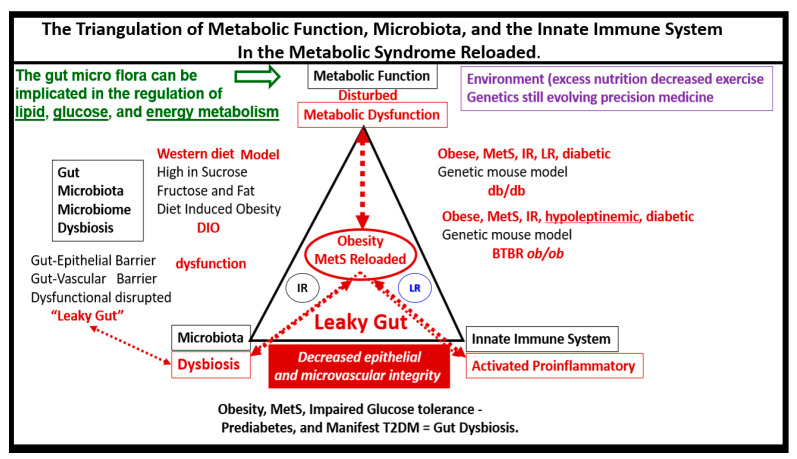
The triangulation of metabolic function, gut microbiota, and the innate immune system. This illustration demonstrates that microbiota dysbiosis, metabolic dysfunction, and an activated proinflammatory innate immune system are bidirectionally associated with obesity and the metabolic syndrome reloaded. Importantly, this triangulation results in a dysfunctional gut-epithelial and gut-vascular barrier that results in a leaky gut to allow increased metainflammation and impaired gut-liver, gut- brain, gut-heart, and gut-kidney homeostasis with resulting impairment of the gut to the brain-heart-kidney axis.

**Figure 19 medicina-59-00561-f019:**
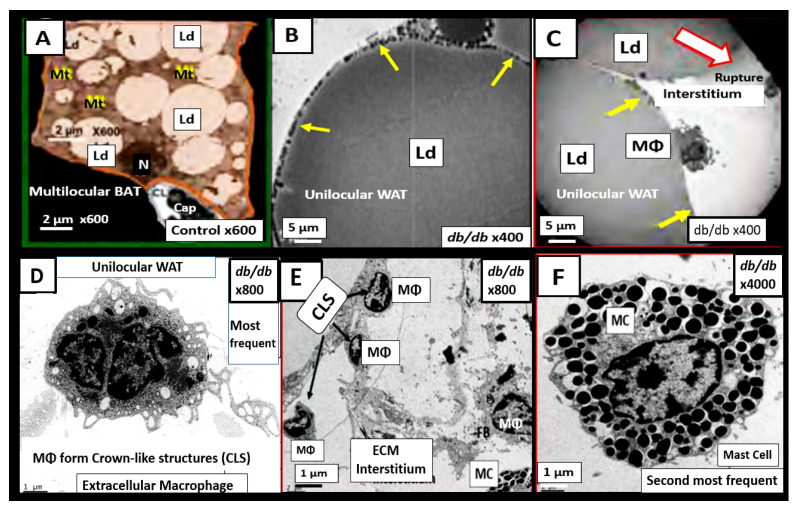
From control multilocular brown adipose tissue (BAT) to unilocular white adipose tissue in the obese, insulin resistant (IR), selective leptin resistant (LR), diabetic *db*/*db* models in aortic perivascular adipose tissue (PVAT) of the *tunica adiposa*. **Panel A** demonstrates the normal multilocular BAT in the PVAT—VAT with multiple lipid droplet(s) (Ld) and electron-dense mitochondria of the descending thoracic aorta in control models. **Panel B** depicts the markedly expanded unilocular WAT with engorgement of triglycerides to form the huge lipid droplet droplets (Ld) with marked thinning of the adipocyte plasma membrane and the compressed mitochondria (yellow arrows). **Panel C** depicts the inflammatory macrophage (MΦ) cytoadherence to the unilocular hypertrophic engorged adipocyte to the point of rupture (red open arrow); note the absence of the mitochondria at the site of rupture. **Panel D** depicts the most prominent inflammatory cell macrophage (MΦ) found within the PVAT of the diabetic *db*/*db* models. **Panel E** depicts the formation of macrophage crown-like structures (CLS) (arrows) in the diabetic *db*/*db* models. **Panel F** depicts the second most prominent inflammatory cell, the mast cell (MC), with its prominent electron-dense secretory granules. These modified images are presented with the permission of CC 4.0 [121,189]. Magnifications and scale bars vary and are included in each image. *ECM = extracellular matrix; Mt = mitochondria; N = nucleus*.

**Figure 20 medicina-59-00561-f020:**
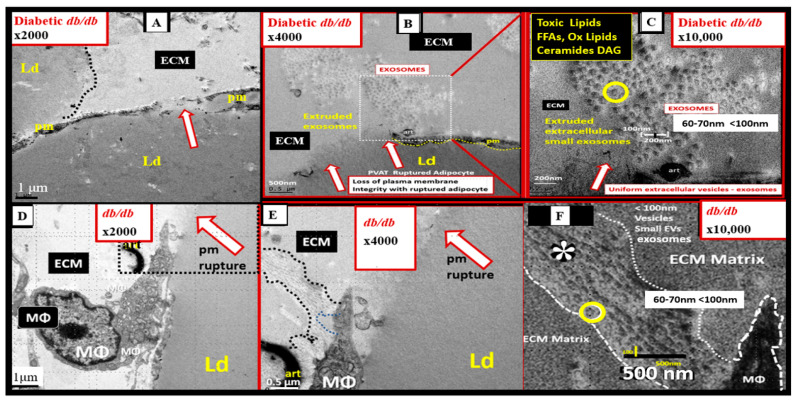
Adipocyte plasma membrane thinning with rupture of plasma membrane (pm) with extrusions of the lipid droplet (Ld) contents and macrophage (MΦ) extrusions of extracellular vesicles (small exosome-like vesicles 60–70 nm in diameter with yellow circles) in the obese, insulin-resistant, leptin-resistant, diabetic *db*/*db* perivascular adipose tissue (PVAT)—visceral adipose tissue (VAT) of the *tunica adiposa* of the descending thoracic aorta. **Panel A** demonstrates the extreme thinning and loss of the plasma membrane integrity (dashed line) and rupture (open red arrow) that is associated with crown-like structures of macrophage cytoadherence. **Panels B** and **C** depict the complete loss of the pm with the rupture and extrusion of the lipid droplet contents into the extracellular matrix (ECM) interstitium of the PVAT—VAT (open red arrows). Importantly, note that these extruded lipids appear as extracellular exosome-like vesicles (encircled) in **Panel C**, in that their diameter is ~60–70 nm, and also note the scale bar that is placed centrally to suggest that their diameter meets the criteria for small exosomes (<100 nm) being extruded from the unilocular ruptured adipocytes in the PVAT. Importantly, the control models that consisted of brown adipose tissue and the models treated with empagliflozin did not demonstrate any crown-like structures (CLS) or ruptured adipocytes as in the diabetic *db*/*db* models. **Panels D**–**F** depict the extrusion of extracellular exosome-like vesicles that are less than 100 nm and are therefore considered to be small macrophage-derived EVexosomes (asterisk and encircled in yellow color) and similar to the morphology of the adipocyte EVexosome-like vesicles in Panel C. Note the red open arrows indicating pm rupture. These modified images are presented with the permission of CC 4.0 [121,189]. Magnification and scale bars vary and are present in each panel.

**Figure 21 medicina-59-00561-f021:**
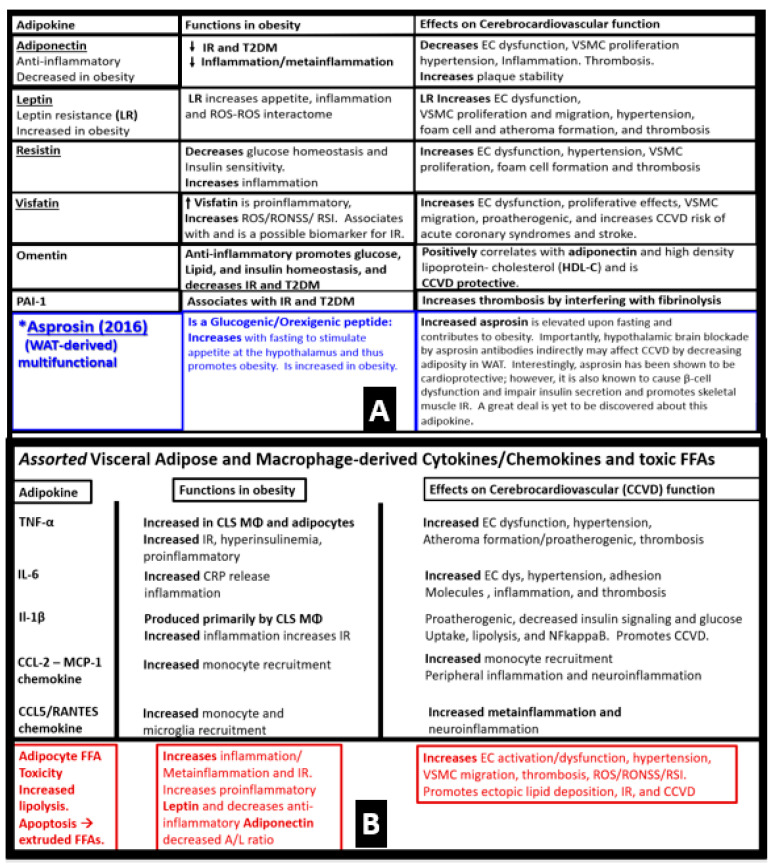
Increased adipokines, assorted cytokines, chemokines, and toxic free fatty acids’ (FFAs) effects on cerebrocardiovascular function and disease. **Panel A (upper panel)** depicts multiple adipokines, their functions in obesity, and effects influencing cerebrocardiovascular disease. Note that the asterisk indicates emerging importance. **Panel B (lower panel)** depicts assorted visceral adipose and adipose-derived macrophage-derived cytokines/chemokines and toxic FFAs. *CCL2 = chemokine ligand 2; CCL5 = chemokine ligand 5; CCVD = cerebrocardiovascular disease; CLS = crown-like structures; EC = endothelial cell; FFA = saturated free fatty acids; IL-1β = interleukin 1-β;IL-6 = interleukin-6; IR = insulin resistance; LR = leptin resistance; MCP-1 = monocyte chemotactic protein-1; MΦ = macrophage; PAI-1 = plasminogen activator inhibitor -1; RANTES = regulated on activation, normal T cell expressed and secreted; ROS = reactive oxygen species; RONSS = reactive oxygen, nitrogen, sulfur species; T2DM = type 2 diabetes mellitus; TNFα = tumor necrosis alpha; VSMC = vascular smooth muscle cell*.

**Figure 22 medicina-59-00561-f022:**
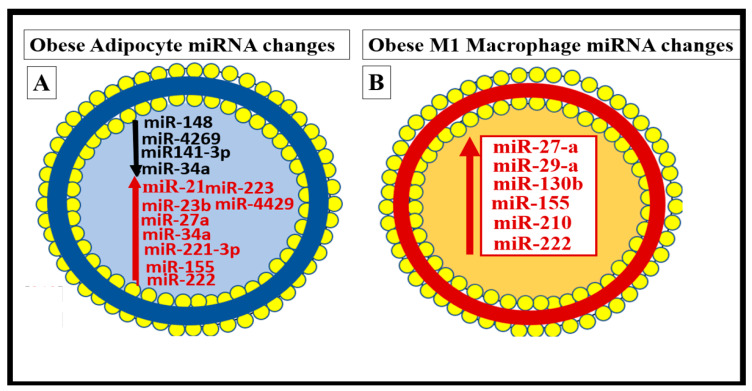
Cell specific miRNAs profiles in adipocytes and activated, M1-like macrophages (MΦs) in the obese PVAT—VAT. **Panel A** depicts the decreased miRNAs (black lettering and black arrow) and the increased miRNAs (red lettering and red arrow) in obese visceral adipose tissue including perivascular adipose tissue (PVAT). **Panel B** depicts the elevated miRNAs (red lettering and red arrow) in the activated M1-like macrophages in the PVAT—VAT [6,204,205,206,207,208,209,210].

**Figure 23 medicina-59-00561-f023:**
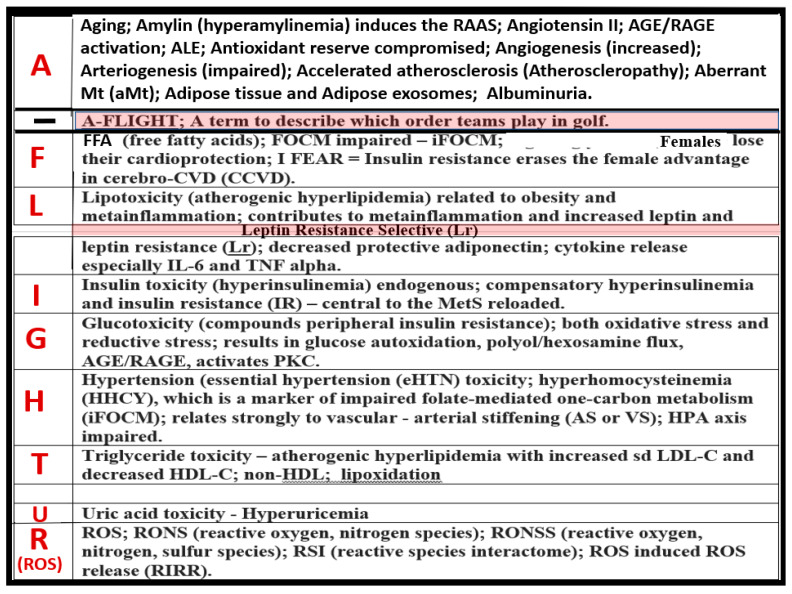
The A-FLIGHT-UR acronym may be utilized as an aid in remembering the multiple risk factors and metabolic toxicities associated with the MetS reloaded. This modified and updated table is provided with permission by CC 4.0 [12]. *AGE = advanced glycation end products; ALE = advance lipoxidation end products; FOCM = folate one-carbon metabolism; HDL-C = high density lipoprotein-cholesterol; HPA = hypothalamic pituitary adrenal; Il-6 = interleukin-6; HDL = high-density lipoprotein; LDL-C = low-density lipoprotein-cholesterol; PKC = protein kinase C; RAGE = receptor or AGE; TNFα = tumor necrosis alpha*.

## Data Availability

Data and materials will be provided upon reasonable request.

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
