# Peer review of "Overview and New Insights into the Metabolic Syndrome: Risk Factors and Emerging Variables in the Development of Type 2 Diabetes and Cerebrocardiovascular Disease"

_medicina, 2023, doi:10.3390/medicina59030561_

Round 1

Reviewer 1 Report

the topic is certainly of great interest but in my opinion the text is too long and inadequately divided into paragraphs. The introduction is redundant although the many historical mentions are interesting. To shorten the text i suggest to remove from line 104 to 142 from line 213 to 219 and from line 299 to line 307

also I do not think it is appropriate the whole part about animal studies (lines 329 to 339; 378-391; 410-414; 486-495; 938-945; 985-988) with related figures and bibliography

from a linguistic point of view too often used the word importantly and started sentences with also

Author Response

RESPONSE TO REVIEWER NUMBER 1

ROUND 1

Dear reviewer number 1

RE: Manuscript identification: medicina-2231969

First, I would like to thank you for your precious time, effort, and knowledge required to review this manuscript.

Comments and Suggestions for Authors

the topic is certainly of great interest but in my opinion the text is too long and inadequately divided into paragraphs. The introduction is redundant although the many historical mentions are interesting. To shorten the text i suggest to remove from line 104 to 142 from line 213 to 219 and from line 299 to line 307

The following lines were removed from the text as follows:

104-142; 213-219; 299-307

Also: Lines 329-339; 378-391; 410-414; 486-485; 938-945; 985-988.

I could not figure out a way to show you these cuts but it will now be very obvious as you read over the revised manuscript

Author agrees about the frequent use of importantly and also and has made a great attempt to reduce the number of these similar introductory words to sentences.

With the multiple above cuts this provided the author a chance to better arrange paragraphs. 

The references have been carefully renumbered and adapted to cuts in this revised submission.

also I do not think it is appropriate the whole part about animal studies (lines 329 to 339; 378-391; 410-414; 486-495; 938-945; 985-988) with related figures and bibliography

from a linguistic point of view too often used the word importantly and started sentences with also

When one has to perform a major revision as in this submission, he does not always know how it will look until all is finished.  I feel your very carefully selected cuts made the manuscript much better and I wish to thank you for this extra time to choose these during your review.

Sincerely, with gratitude,

Melvin R Hayden

University of Missouri School of Medicine

Columbia, Missouri

Reviewer 2 Report

The author provides an extensive review of the factors used to diagnose metabolic syndrome, as well as new molecules and mechanisms associated with this disease. 

The manuscript is thorough, approaching the subject from the history of the disease, but without neglecting the new findings reported.

The manuscript, being so long, is a bit heavy to read, but still readable. 

The document should be improved in its presentation, there are different fonts, size, bold type, spaces, etc.

The references should be revised, for example reference 153 is incomplete. 

The title alludes to new findings, but it is noteworthy that only 10% of the references (approximately) are from the years 2020 to 2023. 

Author Response

RESPONSE TO REVIEWER NUMBER 2

ROUND 1

Dear reviewer number 2

RE: Manuscript identification: medicina-2231969

First, I would like to thank you for your precious time, effort, and knowledge required to review this manuscript.

Comments and Suggestions for Authors

The author provides an extensive review of the factors used to diagnose metabolic syndrome, as well as new molecules and mechanisms associated with this disease. 

The manuscript is thorough, approaching the subject from the history of the disease, but without neglecting the new findings reported.

The manuscript, being so long, is a bit heavy to read, but still readable. 

Author wishes to thank reviewer number 2 for these kind comments.

The document should be improved in its presentation, there are different fonts, size, bold type, spaces, etc.

The Author has carefully gone through the manuscript and all fonts, size and bold type and space and spelling errors have been corrected and I thank you for these kind observations.

The references should be revised, for example reference 153 is incomplete.

All references have been revised and are now complete and all in the same font. 

The title alludes to new findings, but it is noteworthy that only 10% of the references (approximately) are from the years 2020 to 2023.

I agree with this reviewer and since the multiple cuts were made throughout the manuscript I have also made sure all of the references match the change in the text of this revised manuscript.

Thank You,

Respectfully submitted, with gratitude,

Melvin R Hayden

University of Missouri School of Medicine

Columbia, Missouri  

Round 2

Reviewer 1 Report

thanks for your kind answer

i think now the work could be published

Author Response

Thank you for the encouragement

Melvin R Hayden 

Reviewer 2 Report

The author answered my remarks pertinently, you only have to delete the comment of the text editor that you can see in the document presented.

Author Response

Thank you for your encouragement.

Melvin R Hayden